# BMAL2 is a druggable target for ovarian clear cell carcinoma (OCCC)

Grace Y T Tan [ID][1,2], Pei-Yi Lin [ID][2], Li-Tzu Cheng[1,2], Yu-Sheng Tsai Yuan[3], Shih-Han Huang[4], Chen-Hsin Albert Yu[5], Chao-Tsen Chen[3], Peter Chi [ID][4] & Wendy W Hwang-Verslues [ID][2✉]

## Abstract

Ovarian clear cell carcinomas (OCCC), particularly cases that retain wild-type AT-rich interactive domain 1 A (ARID1A) expression (approximately 50% of the OCCC cases), are chemoresistant and lack specific therapies. We identified BMAL2 as a critical OCCC oncogene that promotes tumorigenesis by preventing endogenous DNA damage. BMAL2 depletion altered the expression of genes encoding DNA damage repair proteins, including RAD51, a core enzyme of the homologous recombination (HR) pathway. This led to DNA double-stranded break accumulation, decreased cell viability and reduced tumor growth. This dependence on BMAL2 to maintain DNA integrity and cell viability can be a new route to suppress OCCC. Consistent with this idea, we found that GW833972A, a cannabinoid receptor agonist, bound BMAL2 with high affinity and facilitated its protein degradation. This in turn reduced RAD51 expression, leading to an accumulation of DNA damage, decreased cell viability and reduced OCCC tumor growth. GW833972A is effective by itself at high dose and can also be used at lower dosages to enhance the effectiveness of Poly (ADP-ribose) polymerase inhibitor (PARPi) treatments in BMAL2-expressing OCCC. Together, our findings reveal an essential oncogenic role of BMAL2 and demonstrate that it is an appealing therapeutic target, especially for ARID1A-wt OCCC.

**Keywords** BMAL2; Ovarian Cancer; Ovarian Clear Cell Carcinoma (OCCC); ARID1A; Small Molecule-Induced Protein Degradation
**Subject Categories** Cancer; Urogenital System

## Introduction

Ovarian cancer (OC) is highly metastatic and chemoresistant. Due to its lack of symptoms and rapid metastasis, most OC cases are diagnosed at a late stage (Stage III or IV) when the cancer has spread, and the 5-year survival rate is less than 30%. The current standard treatment for OC is complete cytoreduction in combination with platinum-taxane-based chemotherapy, which can incorporate bevacizumab (Oza et al, 2017) or Poly (ADP-ribose) polymerase inhibitors (PARPi) (Lheureux et al, 2019) as a maintenance therapy after carboplatin and paclitaxel treatment. However, due to complex heterogeneity and chemoresistance, more than 70% of advanced-stage OC patients develop recurrence. Identification of factors that promote these malignant phenotypes is of interest for fundamental OC research and as a means to find new druggable targets.

Five major histological OC subtypes with diverse molecular signatures (Wentzensen et al, 2016) and therapeutic responses (Coward et al, 2015) have been classified: high grade serous ovarian carcinoma (HGSOC), low grade serous (LGSOC), mucinous (MC), endometrioid (EC), and ovarian clear cell carcinoma (OCCC) (Kobel et al, 2014; Koonings et al, 1989; Seidman et al, 2004). HGSOC is the most prevalent subtype, and therefore, has been the most studied. OCCC is the second most prevalent subtype and its incidence rate is rising (Fujiwara et al, 2016; Machida et al, 2019; Park et al, 2017); however, our understanding of OCCC tumorigenic mechanisms is limited. While OCCC may be less aggressive in early stages, it has a significantly worse five-year survival ($p < 0.001$) compared to HGSOC. This is primarily due to its de novo chemoresistance in the advanced stage (Mabuchi et al, 2016). The survival rate for stage III and IV HGSOC patients is about 30%; whereas the survival rate for stage III and IV OCCC patients is ~30% and 0%, respectively (Takano et al, 2006). Numerous treatment strategies have been evaluated for OCCC in clinical trials. However, the overall efficacies are not impressive, and the OCCC subtype-selective effect is low (Ogasawara et al, 2020; Stewart et al, 2023). Unlike HGSOC, most OCCCs (~80%) have wild-type (wt) p53 and low BRCA1/BRCA2 mutation frequency (Lheureux et al, 2019). The most frequent (~50%) alterations in OCCC are mutations in the AT-rich interactive domain 1A gene (ARID1A) that lead to ARID1A deficiency (Lheureux et al, 2019). Treatments that are relatively effective for other OC subtypes, including immune checkpoint blockade, anti-angiogenesis and PARPi, are only partially effective, at best, for OCCC (Stewart et al, 2023). Some of these treatments had some effect to sensitize ARID1A-mutated (ARID1A-mut) OCCC to

[1]Molecular and Cell Biology, Taiwan International Graduate Program, Academia Sinica and National Defense Medical University, Taipei 11490, Taiwan. [2]Genomics Research Center, Academia Sinica, Taipei 115201, Taiwan. [3]Department of Chemistry, National Taiwan University, Taipei 106319, Taiwan. [4]Institute of Biochemical Sciences, National Taiwan University, Taipei 106319, Taiwan. [5]Institute of Molecular Biology, Academia Sinica, Taipei 115201, Taiwan. A previous version has been deposited on bioRxiv: https://doi.org/10.1101/2025.10.21.683700. ✉E-mail: wendyhv@as.edu.tw

DNA-damaging agents; however, the treatments were less effective on ARID1A-wt OCCC (Shen et al, 2018; Yakovlev et al, 2023). Furthermore, it has been reported that ARID1A-wt/TP53-mut OCCC tumors were associated with advanced-stage disease, and these patients had worse overall survival compared to those with ARID1A-mut OCCC tumors (HR = 1.72; 95% CI, 1.06–2.81; $P = 0.03$) (Bolton et al, 2022). Thus, identification of OCCC druggable targets and development of effective targeted therapies, particularly for the ARID1A-wt subset, remains a challenging goal. Detailed characterization of OCCC and how it initiates, progresses and metastasizes will allow more opportunities for OCCC targeted therapeutic development.

Circadian clock genes, including *BMAL1/2*, *CLOCK*, *PER1/2*, and *CRY1/2*, form a complex transcription-translation feedback loop (TTFL) to regulate circadian oscillation and contribute to tissue homeostasis in response to the environment (Ko and Takahashi, 2006). In the circadian system, however, the exact role of BMAL2 (*ARNTL2* and *MOP9*) has not yet been fully understood. It has been shown that transgenic overexpression of BMAL2 can rescue the loss of behavioral rhythms in *Bmal1* knockout mice (Bunger et al, 2000). Nevertheless, such compensatory function is not observed naturally because *Bmal2* expression is decreased in *Bmal1* knockout mice (Sasaki et al, 2009; Shi et al, 2010). In contrast, BMAL2 indirectly suppresses BMAL1 expression via another clock gene, REV-ERBα/β and therefore depletion of BMAL2 may lead to upregulation of BMAL1 in non-cancerous cell lines (Sasaki et al, 2009).

In cancer, these core clock genes are often altered. Among these clock genes, *BMAL2* is the only one upregulated across different types of cancers (Ye et al, 2018), suggesting that BMAL2 may have tumor-promoting functions. Recently, it has been reported that BMAL2 expression may facilitate epithelial-mesenchymal transition (EMT) and is associated with invasion and metastasis phenotypes in breast, colon and lung cancers (Brady et al, 2016; Ha et al, 2016; Lu et al, 2020; Zhang et al, 2022). However, knowledge about the causal relationship between upregulation of BMAL2 and tumorigenesis is limited.

We identified BMAL2 as a critical oncogene in both ARID1A-wt and ARID1A-mut OCCC. BMAL2 depletion resulted in excessive DNA damage, reduced cell proliferation and viability, and consequently inhibited tumor growth. Mechanistically, loss of BMAL2 significantly altered expression of genes in DNA damage repair (DDR) pathways, including homologous recombination (HR)-mediated repair. One of the BMAL2 downstream targets is a core enzyme in the HR pathway, RAD51. Suppression of BMAL2 reduced RAD51 expression and impaired the HR process. Such an impact on DNA integrity revealed a new opportunity for OCCC patients, as BMAL2 depletion significantly reduced OCCC cell viability. Structural-based virtual screening and additional validations found GW833972A, a small molecule previously identified as a selective CB2 cannabinoid receptor agonist, as a BMAL2 inhibitor. GW833972A facilitated BMAL2 protein degradation, which consequently reduced RAD51 and other DDR genes to increase DNA damage and decrease cell viability. Xenograft models using ARID1A-wt OCCC cells further demonstrated that GW833972A treatment alone at higher doses or in combination with PARPi at low doses can inhibit tumor growth. Together, our findings demonstrated that BMAL2 is a critical oncogene which prevents endogenous DNA damage in OCCC and can be a

biomarker as well as a key therapeutic target, particularly for ARID1A-wt OCCC.

# Results

## BMAL2 upregulation is observed in OC, including the OCCC subtype

*BMAL2* expression is extremely low in normal ovarian tissues (Fig. 1A,B). However, close to 50% of OC (TCGA-OV) cases in the Cancer Genomics Atlas (TCGA) database have increased copy number of *BMAL2* (Fig. 1C). This is substantially higher than other circadian clock genes and suggests that these tumors have increased BMAL2 expression. Kaplan–Meier (K–M) analysis (Gyorffy, 2023) further showed that OC patients with high *BMAL2* expression tumors have significantly worse overall survival (OS) and reduced progression-free survival (PFS) compared to patients with low *BMAL2* expression tumors (Fig. 1D,E), indicating a positive correlation between BMAL2 expression and OC malignancy. To determine whether BMAL2 is differentially expressed in OC histological subtypes, the CSIOVDB microarray gene expression database was used, and the results showed that most OCCC cases have significantly higher *BMAL2* expression compared to SOC (HGSOC and LGSOC) and EC subtypes (Fig. 1F). OCCC is generally considered to be derived from endometrial tissue. However, the acquisition of paraffin-embedded ectopic endometrial tissue is challenging. Thus, we used normal endometrioid epithelium (NEE) samples instead to perform immunohistochemistry (IHC) to compare the BMAL2 level with OCCC primary specimens. The results showed that BMAL2 was exclusively expressed in OCCC cells, not in the adjacent non-tumor cells (NT) or NEE (Fig. 1G), indicating that BMAL2 has potential prognostic value and oncogenic roles in OCCC.

## BMAL2 depletion inhibits tumorigenic ability in OCCC cells

Approximately half of OCCC cases harbor mutated ARID1A, while the other half expresses wt ARID1A (Lheureux et al, 2019). Since ARID1A is an accessory subunit of the SWI/SNF chromatin remodeling complex, transcriptomic profiles between these two subsets of OCCC differ from each other, and, consequently, the mechanisms driving tumorigenic ability and drug response can also be different. To investigate whether ARID1A status may influence BMAL2 levels, we examined a panel of OCCC cell lines and found that BMAL2 protein level was higher in most of the OCCC cells regardless of their ARID1A status when compared to primary human endometrial stromal cells (HESC) and human ovarian surface epithelial cells (HOEC) (Fig. 2A). OCCC cell lines with high BMAL2 level were used for loss-of-function assays where BMAL2 was depleted with shRNA (Fig. 2B). Importantly, BMAL2 depletion inhibited cell proliferation (Fig. 2C), reduced DNA synthesis (Figs. 2D and EV1A), and decreased mitosis (Figs. 2E and EV1B) in both ARID1A-wt and ARID1A-mut cell lines. BMAL2 knockdown also inhibited clonogenic ability in vitro (Figs. 2F and EV1C). Subcutaneous xenografts using NUDE mice injected with shCtrl or shBMAL2 (clone #1) lentiviral transduced ARID1A-wt OCCC cell lines, ES-2 and JHOC5, and ARID1A-mut OCCC cell line, OVISE,

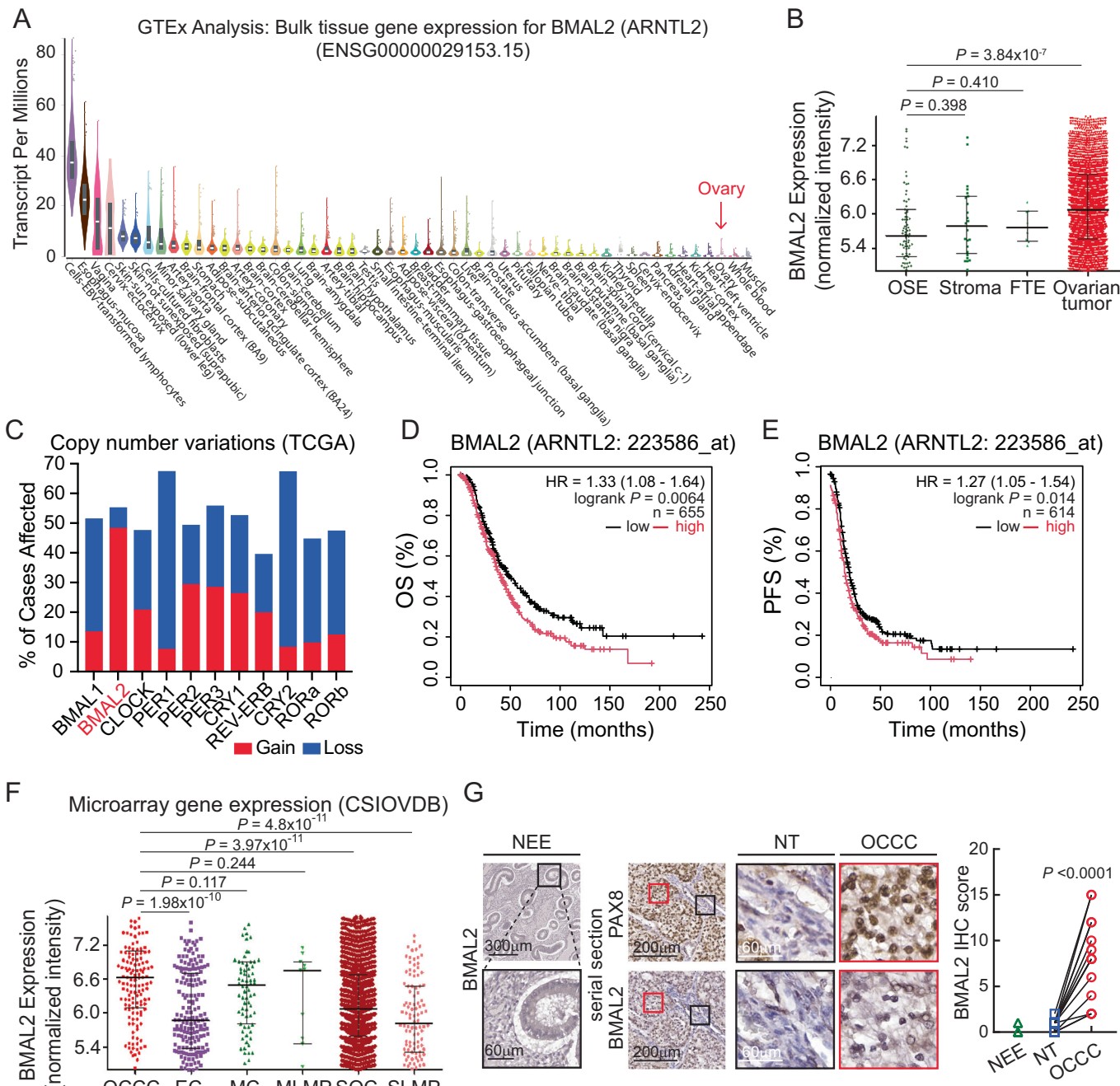

showed that BMAL2 depletion reduced tumor growth (Figs. 2G–I and EV1D). Immunohistochemistry (IHC) analysis further confirmed that shBMAL2 tumors had decreased proliferating cells, indicated by reduced Ki67-positive cells (Figs. 2J–L and EV1E). Together, these results suggested that BMAL2 plays an important role in OCCC tumorigenesis.

We also examined publicly available datasets to investigate whether BMAL2 expression is prognostically related to ARID1A status of OCCC. We reanalyzed raw data from the study of Bolton et al (2022) and extracted 164 cases that have information on ARID1A status, gene expression and clinical outcome. ARID1A-wt

patients ($n = 65$), regardless of their p53 status, tended to have worse overall survival than ARID1A-mut patients ($n = 99$), although the difference between these two groups was not statistically significant ($P = 0.26$) (Fig. EV2A). We then further ranked these tumors by BMAL2 level, and compared the first (BMAL2-high) and fourth (BMAL2-low) quartiles both within and between ARID1A-wt ($n = 17$ for each group) and ARID1A-mut ($n = 25$ for each group) cases. No statistical significance was observed ($P = 0.56$) (Fig. EV2B). While the sample size of this dataset was small, the results further indicated that the BMAL2 function may be independent of ARID1A status.

◀ **Figure 1. High BMAL2 expression is observed in OC, including OCCC, and is correlated with poor clinical outcomes.**

(A) BMAL2 (ARNTL2) gene expression in normal tissues. The data were obtained from the GTEx Portal (dbGaP Accession phs000424.v8.p2). Expression values are shown as Transcripts Per Million (TPM). Box plots show the median and 25th and 75th percentiles. Points are displayed as outliers if they are above or below 1.5 times the interquartile range. Sample size of each tissue and median TPM are listed in Dataset EV2. (B) BMAL2 gene expression in normal ovarian surface epithelium (OSE) ($n = 91$), stroma ($n = 25$), fallopian tube epithelium (FTE) ($n = 8$) and ovarian tumor ($n = 3139$) was analyzed using the CSIOVDB database. Data were shown as median with 25th and 75th quartiles. The P value was determined by Mann–Whitney test. (C) Copy number variation of BMAL2 in OC (TCGA-OV) cases. (D) Kaplan–Meier overall survival (OS) analysis of OC patients grouped by BMAL2 expression. BMAL2-high group ($n = 372$) is indicated by the red line; BMAL2-low group ($n = 283$) is indicated by the black line. $P = 0.0064$. The P value was determined by the log-rank test. (E) Kaplan–Meier progression-free survival (PFS) analysis of OC patients grouped by BMAL2 expression. BMAL2-high group ($n = 215$) is indicated by the red line; BMAL2-low group ($n = 399$) is indicated by the black line. $P = 0.014$. The P value was determined by the log-rank test. (F) BMAL2 gene expression in OC histological subtypes analyzed using the CSIOVDB database. Data were shown as median with 25th and 75th quartiles. The P value was determined by Mann–Whitney test. OCCC, clear cell carcinoma ($n = 132$); EC endometrioid carcinoma ($n = 167$); MC mucinous carcinoma ($n = 70$); SOC serous carcinoma ($n = 2200$), LMP low malignant potential (MLMP, $n = 10$; SLMP, $n = 96$). (G) (Left panel) Representative images of BMAL2 IHC staining using normal endometrioid epithelium (NEE). The black box indicates an enlarged region. (Scale bars, 300 or 60 μm as shown in the images.) (Middle panel) Representative images of PAX8, a reliable clinical marker for OCCC (Chai et al, 2017; Gorbokon et al, 2024), and BMAL2 IHC staining using serial tumor sections from OCCC patients. Red boxes indicate enlarged tumor regions. Black boxes indicate enlarged non-tumor (NT) regions. (Scale bars, 200 or 60 μm as shown in the images.) (Right panel) Quantification and comparison of BMAL2 level in six NEE samples, and NT and OCCC tumor regions of 14 OCCC patient samples. The BMAL2 IHC score was determined by calculating the product of the percentage of positive tumor cells and the intensity of the staining. The tumor percentage score is based on a semi-quantitative five-category grading system: 0 (no tumor cell), 1 (1–10%), 2 (11–25%), 3 (25–50%), 4 (51–75%), and 5 (>75%). The staining intensity score is based on a semi-quantitative four-category grading system: 0 (no staining), 1 (weak staining), 2 (moderate staining), and 3 (strong staining). Significant difference is based on a paired t-test. Source data are available online for this figure.

## Loss of BMAL2 increases endogenous DNA damage in OCCC cells

To understand how BMAL2 depletion inhibited OCCC tumorigenesis, RNA-seq analysis was performed using ARID1A-wt OVCA429 and ARID1A-mut OVISE cells without or with BMAL2 depletion. KEGG pathway enrichment analysis indicated that in both cell lines, the differentially expressed genes (DEG) were enriched in pathways including DNA damage repair (DDR), homologous recombination (HR), cell cycle and cellular senescence (Fig. 3A). These pathways are functionally interconnected as DDR and HR are activated upon DNA damage. And if the damage is too severe or cannot be fixed, the cell cycle can be arrested, leading to cellular senescence.

To validate whether BMAL2 depletion increased DNA damage, the physical state of DNA and biological response to DNA double-strand breaks were evaluated using alkaline comet assay (Olive and Banath, 2006) and γH2AX (Burma et al, 2001) staining, respectively. The comet assay showed a significant increase in tail moment in the BMAL2-depleted cells (Figs. 3B and EV3A), suggesting that loss of BMAL2 leads to DNA damage. BMAL2 knockdown also elevated γH2AX level and foci numbers (Figs. 3C and EV3B,C), indicating that BMAL2 depletion significantly increased DNA double-strand breaks. It is worth mentioning that the DNA damage resulting from BMAL2 depletion was only observed in OCCC cells, not in serous OC subtypes. Among commonly used serous OC cell lines, HEYA8 and KURAMOCHI expressed a high level of BMAL2 compared to HOEC cells and other serous OC cell lines (Fig. EV3D). BMAL2 knockdown did not lead to an increase in γH2AX foci in HEYA8 and KURAMOCHI cells (Fig. EV3E,F), suggesting that BMAL2-mediated DNA integrity maintenance is an OCCC subtype-specific regulatory mechanism.

Importantly, most of the genes in the HR pathway were downregulated when BMAL2 was depleted (Fig. EV4A,B; Dataset EV1). In particular, expression of RAD51, a crucial protein responsible for the core steps of HR in eukaryotes (Li and Heyer, 2008), was significantly downregulated (Fig. EV4C,D). Since

BMAL2 is a transcriptional factor that regulates its downstream gene expression via binding to E-boxes, a regulatory motif of DNA with the consensus sequence 5'-CANNTG-3' (Sasaki et al, 2009), we further investigate whether BMAL2 directly upregulates RAD51 transcription. We used The Eukaryotic Promoter Database (EPD) (Dreos et al, 2015) to examine the RAD51 proximal promoter and found three E-box containing motifs ($-983$ to $-978$, $-802$ to $797$, $-209$ to $-202$) (Fig. EV4E). Chromatin immunoprecipitation (ChIP) further confirmed the binding of BMAL2 to these E-box regions (Fig. EV4F), indicative of activation of RAD51 transcription when BMAL2 was present.

Since deficiency in RAD51 is known to impair HR and lead to DNA damage accumulation (Li and Heyer, 2008), and BMAL2 depletion resulted in downregulation of RAD51 (Fig. EV4C,D), we employed an HR activity reporter assay (Lee et al, 2023) to determine whether BMAL2 depletion led to HR deficiency and therefore resulted in DNA damage accumulation. As a comparison, we treated the cells with B02, a RAD51-specific inhibitor (Fig. EV4G). BMAL2 depletion greatly inhibited HR, even more so than B02 treatment (Fig. EV4G). Together, these results indicate that RAD51 transcriptional regulation is one of the multifaceted functions of BMAL2 to maintain HR and prevent DNA damage in OCCC cells.

## Depletion of BMAL2 alone or with PARP inhibitor (PARPi) can inhibit the tumorigenic ability of ARID1A-wt OCCC cells

Olaparib, a PARP inhibitor (PARPi), has been approved in the United States and Europe as a maintenance therapy for platinum-sensitive relapsed BRCA-mutated OC that displays defects in the HR repair pathway. It has been reported that 6.25 μM Olaparib can kill half of the population ($IC_{50}$) of primary OC cells with HR deficiency (Lee et al, 2023). We examined the sensitivity of OCCC cells to Olaparib (dose range 0 to 10 μM) and found that ARID1A-mut OCCC cells were more sensitive to PARPi than ARID1A-wt OCCC cells (Fig. EV5). The Olaparib $IC_{50}$ was below 2.5 μM for OVISE (ARID1A-mut) cells, but was much higher for ARID1A-wt OCCC cells.

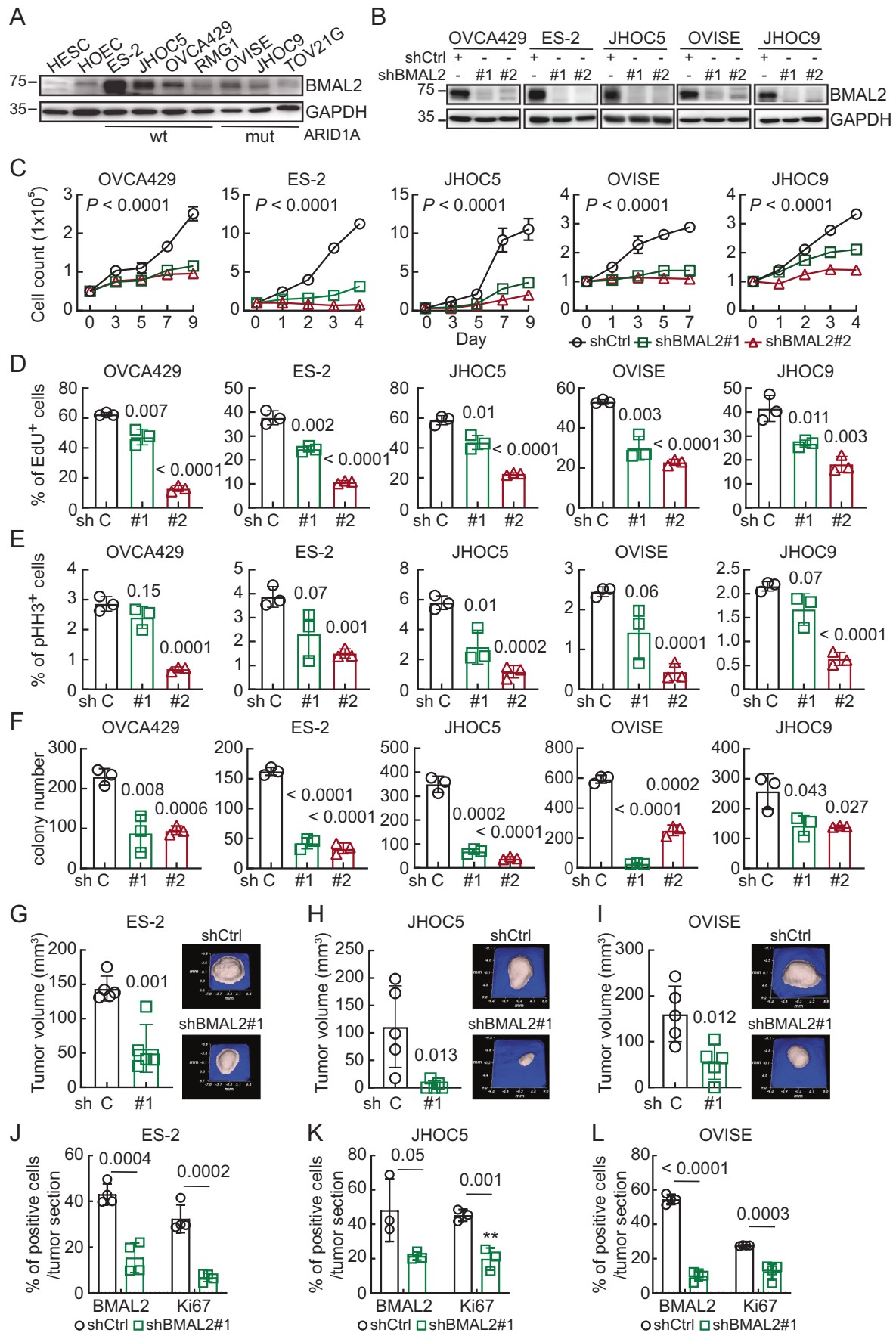

◄ **Figure 2. BMAL2 depletion inhibits tumorigenic ability in OCCC cells.**

(A) IB of BMAL2 protein expression with GAPDH as a loading control. Blots shown are from one representative experiment of three replicates. (B) IB of BMAL2 protein expression with GAPDH as a loading control in BMAL2-depleted (shBMAL2#1 or #2) OCCC cells. Blots shown are from one representative experiment of three replicates. (C) Cell growth assays using control (shCtrl) or BMAL2-depleted (shBMAL2) OCCC cells. Data were shown as mean ± SD with $P$ value based on two-way ANOVA test ($n = 3$). Data shown are from one representative experiment of three replicates. (D) EdU staining assays using shCtrl or shBMAL2 OCCC cells. Data are shown as mean ± SD with $P$ value based on unpaired $t$-test ($n = 3$). Data shown are from one representative experiment of three replicates. (E) Phospho-histone H3 (pHH3) staining assays using shCtrl or shBMAL2 OCCC cells. Data were shown as mean ± SD with $P$ value based on unpaired $t$-test ($n = 3$). Data shown are from one representative experiment of three replicates. (F) Clonogenic assays using shCtrl or shBMAL2 OCCC cells. Data are shown as mean ± SD with $P$ value based on unpaired $t$-test ($n = 3$). Data shown are from one representative experiment of three replicates. (G–I) Xenograft models in NUDE mice using shCtrl or shBMAL2 ES-2 (G), JHOC5 (H), and OVISE (I) cells. Five mice were used for each group. Data were means ± SD, significant difference is based on unpaired $t$-test of the tumor volume 3, 4, and 8 weeks after subcutaneous injection, respectively. Representative 3D ultrasound tumor images are shown here, and all individual tumor images are presented in Fig. EV1D. (J–L) Quantification and comparison of the percentage of BMAL2 and Ki67 expressing cells detected by IHC in shCtrl or shBMAL2 ES-2 (J), JHOC5 (K), and OVISE (L) derived tumors. Three to four tissue sections from each of the representative tumors shown in (G–I) were used (ES-2, $n = 4$; JHOC5, $n = 3$; OVISE, $n = 4$) for staining and quantification. Data were shown as mean ± SD with $P$ value based on an unpaired $t$-test. Source data are available online for this figure.

Since BMAL2 depletion impaired HR in OCCC, we reasoned that targeting BMAL2 may enhance PARPi sensitivity in ARID1A-wt OCCC cells and inhibit their tumorigenic ability. In the shCtrl OCCC cells, 2.5 μM Olaparib reduced colony numbers by 30% in ES-2 cells (Fig. 3D) and 40-45% in JHOC5 cells (Fig. 3E). Importantly, depletion of BMAL2 was itself able to achieve a much more pronounced inhibitory effect (50% reduction) than Olaparib in ES-2 cells (Fig. 3D). When BMAL2-depletion was combined with Olaparib, the colony-forming ability of ARID1A-wt OCCC cells was further reduced (Fig. 3D,E). These results indicated that targeting BMAL2, alone or with PARPi, can be an effective therapeutic strategy for ARID1A-wt OCCC.

## GW833972A can be used to target the BMAL2 protein

Transcription factors are traditionally thought to be undruggable; however, recent development of small-molecule modulators has suggested that targeting transcription factors can be feasible (Bushweller, 2019). We utilized a structure-based virtual screening to evaluate potential drugs that can be used to target BMAL2. Currently, the full-length BMAL2 protein structure has not been solved. The C-terminal PAS2 domain (N360-E477) of BMAL2 is the only available 3D structure resolved by the NMR technique [Protein Database (PDB) Entry- 2KDK, Fig. EV6A]. Thus, we used the PAS2 domain structure and the Alphafold protein structure database-predicted BMAL2 full-length 3D structure to screen for potential binding pockets. A surface binding pocket containing E406, Y407, F408, H409, Q410, R438, A439, and F444 residues in the PAS2 domain was found and used as a docking area to perform virtual screening (Fig. EV6B). Based on the docking scores, drug properties, and binding mode analysis, 119 high-affinity compounds (FRED Chemgauss4 score between −9.3 and −7.8) were identified from the chemical databases, including more than 60,000 compounds. Within the 119 candidates, the top nine bioactive compounds with a low $K_d$ value were then selected for further experiments. The impact of each selected compound on the BMAL2 protein level was evaluated using ARID1A-wt ES-2 cells (Fig. EV6C). GW833972A, a cannabinoid receptor 2 (CB2) selective agonist (Belvisi et al, 2008), was the most effective compound to reduce BMAL2 expression at the lowest dose (5 μM, Fig. EV6C). GW833972A was predicted to interact with residue Q410 in the BMAL2 binding pocket via hydrogen bonding (Figs. 4A and EV6D). Importantly, GW833972A did not down-regulate BMAL2 mRNA expression in any of the five OCCC cell lines tested (Fig. EV6E), but substantially reduced its protein level in all five cell lines (Fig. 4B). To determine whether GW833972A binds to BMAL2 protein, cellular thermal shift assay (CETSA) and drug affinity responsive target stability (DARTS) analysis were performed (Jafari et al, 2014; Lomenick et al, 2009). The thermal stability of the BMAL2 protein was significantly increased (Fig. 4C), and the BMAL2 protein was more resistant to enzymatic digestion when GW833972A was present (Fig. 4D), indicating target engagement of GW833972A with the BMAL2 protein. Cycloheximide (CHX) chase assay was further performed to evaluate BMAL2 protein half-life in response to GW833972A treatment. BMAL2 protein was less stable when the cells were treated with GW833972A (Fig. 4E,F). Furthermore, inhibition of the proteasome using MG132 prevented BMAL2 protein degradation (Fig. 4G,H). Together, results from these biochemical assays suggested that GW833972A physically interacted with BMAL2, as predicted by the structure-based virtual modeling (Fig. 4A), and facilitated BMAL2 protein degradation via the proteasome system.

## Targeting BMAL2 by GW833972A provides a therapeutic opportunity for OCCC

All five OCCC cell lines also showed reduced RAD51 level after GW833972A treatment (Fig. 4B), indicative of decreased BMAL2 levels. GW833972A treatment also increased γH2AX foci (Fig. 5A), inhibited cell proliferation (Fig. 5B), decreased colony-forming ability (Fig. 5C) and reduced cell viability (Fig. 5D) in a dose-dependent manner. These phenotypes were consistent with the effect of decreasing BMAL2 protein levels by shRNA (Figs. 2 and 3).

We also tested the combined treatment of GW833972A and Olaparib. A low level of GW833972A (10 μM) significantly enhanced Olaparib sensitivity in ES-2 and JHOC5 cells, with cell viability decreased from 75 to 40% (Fig. 5E,F). Higher dose of GW833972A (20 μM) alone reduced cell viability to 40%, and the combination of this high dose of GW833972A with Olaparib did not further suppress cell viability (Fig. 5E,F). The combinatory effect of GW833972A and Olaparib was quantitatively evaluated using the Bliss independence model calculated by SynergyFinder 3.0 (Ianevski et al, 2022). In ES-2 cells, the overall synergy score was 13.48, and the score of the most synergistic area (1.25–5 μM Olaparib and 5–20 μM GW833972A) was 24.45, indicating a strong synergistic effect when low doses of GW833972A and Olaparib

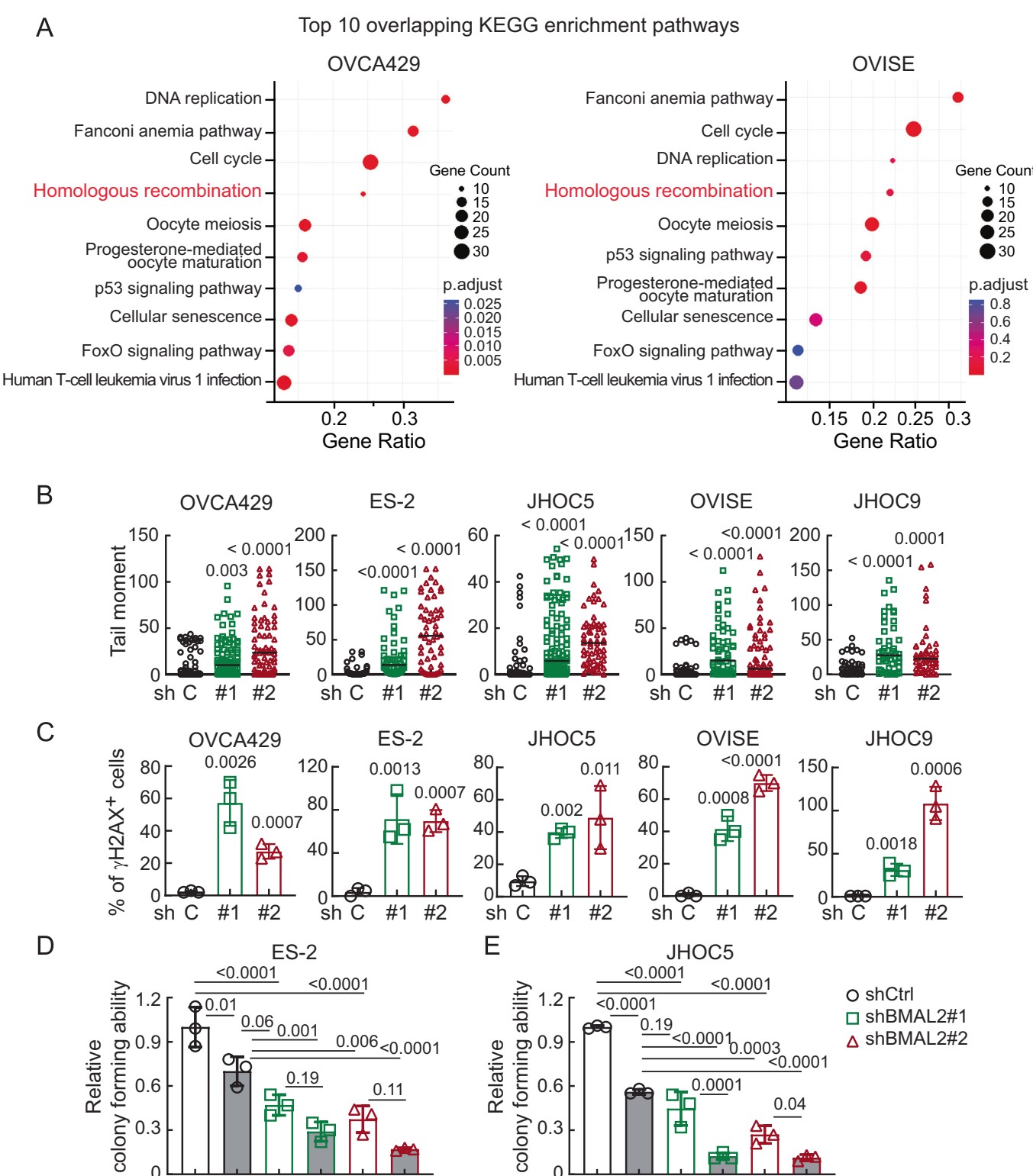

**A** Top 10 overlapping KEGG enrichment pathways

were combined (Fig. 5G). In JHOC5 cells, the overall synergy score was 7.53, and the score of the most synergistic area (5–10 μM Olaparib and 10–40 μM GW833972A) was 10.26, indicating a moderate synergistic effect when both drugs were used (Fig. 5H). Together, these results indicate that GW833972A is effective by

itself at high doses and can also be used at lower dosages to enhance the effectiveness of PARPi treatments in BMAL2-expressing OCCC.

It is known that BMAL2 and BMAL1 share the same overall domain architecture. Since BMAL1 is essential for maintaining

**Figure 3. BMAL2 depletion increases endogenous DNA damage in OCCC cells.**

(A) KEGG Enrichment Analysis of downregulated genes in BMAL2-depleted OVCA429 and OVISE cells. KEGG pathway enrichment analysis was performed using over-representation analysis (ORA) based on a hypergeometric test. P values were adjusted for multiple comparisons using the Benjamini–Hochberg false discovery rate (FDR) method. The top ten overlapping pathways, determined by their adjusted P value, are shown. (B) Tail moment of comet assays using shCtrl or shBMAL2 OCCC cells without exposure to DNA damage agents. At least 50 cells per group were measured. Significant differences are based on an unpaired t-test. Data shown are from one representative experiment of three replicates. (C) γH2AX staining assays using shCtrl or shBMAL2 OCCC cells without exposure to DNA damage agents. Data were shown as mean ± SD with P value based on unpaired t-test (n = 3). Data shown are from one representative experiment of three replicates. (D, E) Clonogenic assays using shCtrl or shBMAL2 ES-2 (D) and JHOC5 (E) cells without or with 2.5 μM Olaparib treatment. Data were shown as mean ± SD with P value based on one-way ANOVA test (n = 3). Data shown are from one representative experiment of three replicates. Source data are available online for this figure.

circadian rhythm and regulating physiological processes, we evaluated whether GW833972A also targets BMAL1, which could lead to adverse effects in normal tissues. The structure of the BMAL1 PAS2 (a.k.a. PAS-B) domain in complex with a core circadian modulator (CCM) was recently resolved by X-ray diffraction (Protein Data Bank: 8RW8) (Pu et al, 2025). Thus, this CCM binding pocket was used for docking analysis to evaluate whether GW833972A could bind to BMAL1 (Fig. EV7A). In contrast to the surface-exposed binding pocket of BMAL2, the CCM binding site of BMAL1 is an internal cavity (Pu et al, 2025). Also, the BMAL1 and BMAL2 residues that are in positions most likely to contact GW833972A have different physicochemical properties. The residues of the BMAL1 binding pocket are hydrophobic (Fig. EV7A), while the residues of the BMAL2 binding pocket are hydrophilic or uncharged (Fig. 4A). These differences in structure and chemical properties could make it difficult for GW833972A to access the BMAL1 binding pocket. Consistent with these structural differences, treating BMAL1-expressing cells, such as ES-2 OCCC cells and WI38 human fetal lung fibroblasts, with GW833972A did not reduce BMAL1 protein level (Fig. EV7B). Together, these results indicate that GW833972A does not bind to or cause degradation of BMAL1 protein.

To further determine whether GW833972A is toxic to normal cells, WI38 fibroblasts were used. No significant decrease in cell viability or DNA damage was observed when WI38 cells were treated with 20 μM GW833972A (Fig. EV7C,D). Moreover, GW833972A had a much higher $IC_{50}$ for WI38 fibroblasts ($IC_{50}$: 133.6 μM) than for ES-2 OCCC cells ($IC_{50}$: 20.2 μM) (Fig. EV7E). The low toxicity of GW833972A in WI38 fibroblasts was likely due to their extremely low BMAL2 expression (Fig. EV7F). Similarly, in low BMAL2 OCCC cells, no further BMAL2 depletion was detected upon GW833972A treatment (Fig. EV7G). In TOV21G cells, no significant reduction in cell viability was found when 20 μM GW833972A was used (Fig. EV7H). RMG1 cells had a decrease in cell viability when treated with 20 μM GW833972A, but no significant effect was observed when a lower dose was used (Fig. EV7I). These results further demonstrate the drug-target specificity between GW833972A and BMAL2. Since depletion of BMAL2 only inhibited growth and viability of cells with high BMAL2, and BMAL2 expression is low in most normal tissues (Fig. 1A), these observations suggest that GW833972A can effectively target BMAL2-expressing cancer cells with little adverse effects on normal cells.

To validate these cell-based observations in vivo, subcutaneous xenograft models using ES-2 and JHOC5 cells were performed (Figs. 6 and EV8). When tumors reached 50 mm³, mice were intraperitoneally injected with either vehicle or 10 mg/kg GW833972A daily for five days a week over the course of 2 weeks

(Fig. EV8A). The experiments were terminated when tumor-induced skin breakage occurred in the vehicle control group. In both mouse models, GW833972A treatment significantly inhibited tumor growth without body weight changes (Figs. 6A,B and EV8B,C). No abnormalities were observed in the liver, kidney and spleen in these mice (Fig. EV8D,EV8E), suggesting low adverse side effects of GW833972A. These tumors showed reduced BMAL2 and RAD51 positive cells, increased DNA double-stranded breaks (γH2AX) and decreased cell proliferation (Ki67) compared to the vehicle-treated control (Fig. 6C–F). We then tested the combination effect of GW833972A and Olaparib in an ES-2 xenograft model (Fig. 6G). When tumors reached 50 mm³, mice were intraperitoneally injected daily with either vehicle, 10 mg/kg Olaparib, 10 mg/kg or 20 mg/kg GW833972A, or GW833972A-Olaparib combination until the experiment was terminated. Similar to the cell-based experiment (Fig. 5), when GW833972A or Olaparib was applied individually, tumor growth was inhibited. When 10 mg/kg GW833972A was used in combination with Olaparib, the tumor inhibition effect was further enhanced, but less than the sum of the individual effects. Monotherapy using 20 mg/kg GW833972A can achieve a more significant effect than 10 mg/kg GW833972A and Olaparib combined therapy. When the dose of GW833972A was increased, the combination effect with Olaparib was still detected, but was smaller. Importantly, GW833972A treatment as monotherapy or in combination with Olaparib did not affect mouse body weight (Fig. 6H) and did not cause abnormalities in the liver, kidney and spleen in these mice (Fig. EV8F). Together, the data showed that targeting BMAL2 by GW833972A provides a therapeutic opportunity for OCCC.

## Discussion

Based on our findings that BMAL2 expression was extremely low in non-tumor cells in ovarian tissues but upregulated in OCCC cells, and our observations that targeting BMAL2 led to excessive DNA damage, reduced cell viability and inhibited tumor growth regardless of the ARID1A status, we propose that BMAL2 can serve as a biomarker as well as a key therapeutic target for OCCC. Moreover, since BMAL2 regulates DNA damage repair pathways to prevent DNA damage, its upregulation in OCCC may also be an important contributing factor to the high chemoresistance of this refractory disease. Our experiments have demonstrated that targeting BMAL2 allows more effective suppression of OCCC growth, particularly ARID1A-wt OCCC, for which current treatment options are severely limited.

While many clock genes, including BMAL1, play roles in DNA damage repair to maintain genome stability (Sancar et al, 2010),

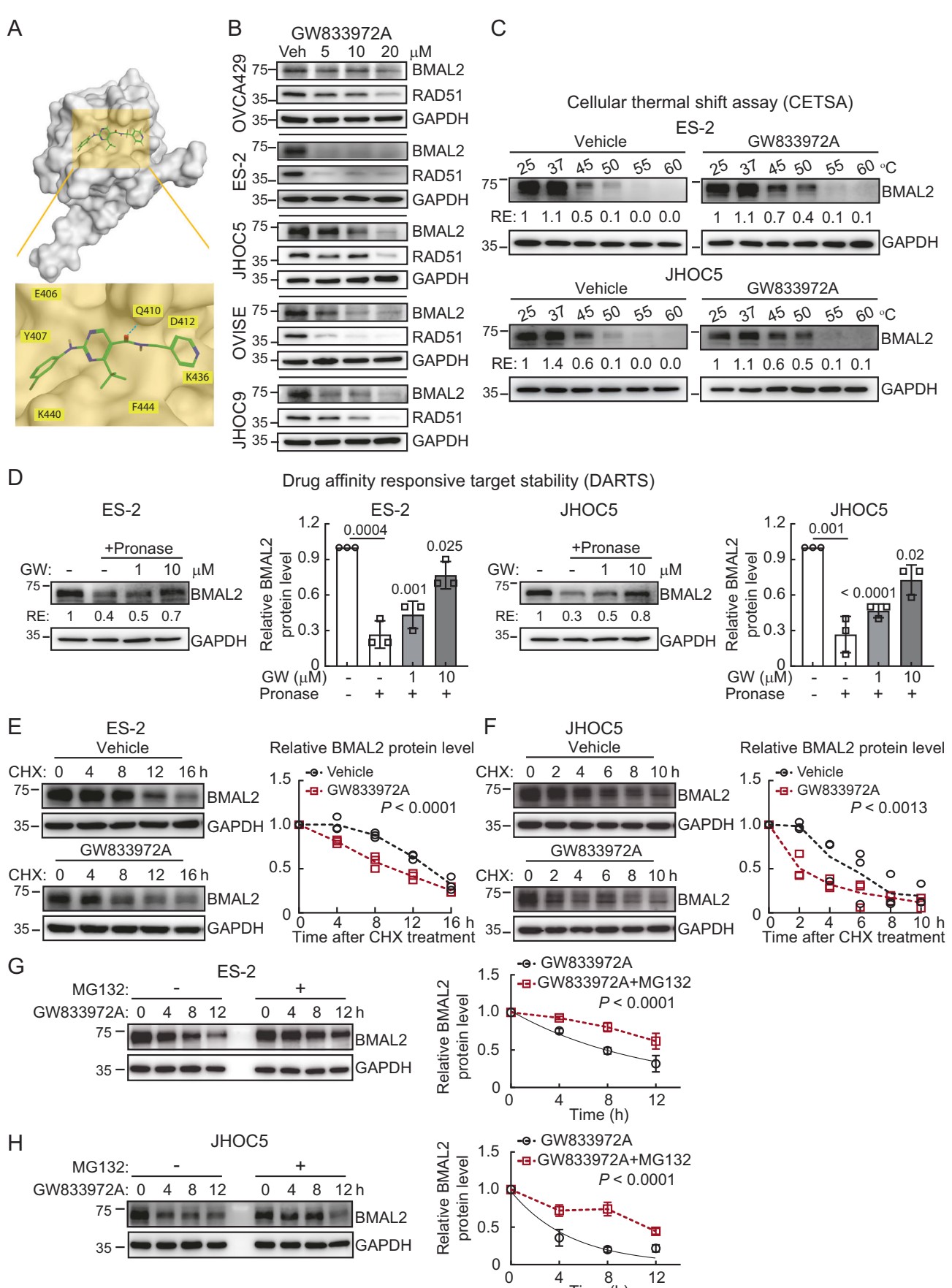

◀ **Figure 4. GW833972A targets the BMAL2 protein for degradation.**

(A) Binding mode of GW833972A to human BMAL2. BMAL2-compound complex, in which the BMAL2 PAS2 domain is shown as a molecular surface, and GW833972A is shown in green. The interaction of GW833972A with the contact residues of BMAL2 are shown. Residue Q410 forms a hydrogen bond with GW833972A. (B) IB of BMAL2 and RAD51 protein expression with GAPDH as a loading control in OCCC cells treated with vehicle, 5, 10, or 20 μM GW833972A. Blots shown are from one representative experiment of three replicates. (C) Cellular thermal shift assay (CETSA) using ES-2 and JHOC5 cells treated with vehicle or 20 μM GW833972A. The level of BMAL2 was determined using IB analysis. GAPDH was used as a loading control. Blots shown are from one representative experiment of three replicates. RE relative expression. (D) Drug affinity responsive target stability (DARTS) assay using ES-2 and JHOC5 cells treated with vehicle, 1 or 10 μM GW833972A, without or with 0.2 μg/ml Pronase. The level of BMAL2 was determined using IB analysis. GAPDH was used as a loading control. Blots shown are from one representative experiment of three replicates, and relative BMAL2 protein levels from three IB analyses were quantified. Data were means ± SD with $P$ value based on an unpaired $t$-test. (E) Time course assay using cycloheximide (CHX) treated ES-2 cells without or with 20 μM GW833972A treatment. The level of BMAL2 was determined using IB analysis. GAPDH was used as a loading control. Relative BMAL2 protein levels from three IB analyses were quantified, and nonlinear regression (curve fit) analysis was used to test for significant differences between the degradation curve of the vehicle control and GW833972A-treated group ($P$ value is indicated). (F) Time course assay using CHX-treated JHOC5 cells without or with 20 μM GW833972A treatment. The level of BMAL2 was determined using IB analysis. GAPDH was used as a loading control. Data formatting is as described for (E). (G, H) Time course assay using 20 μM cycloheximide (CHX) and 10 μM MG132 treated ES-2 (G) and JHOC5 (H) cells without or with 20 μM GW833972A treatment. The level of BMAL2 was determined using IB analysis. GAPDH was used as a loading control. Relative BMAL2 protein levels from three IB analyses were quantified. Data were shown as mean ± SD ($n = 3$). Nonlinear regression (curve fit) analysis was used to test for significant differences between the degradation curve of the vehicle control and GW833972A-treated group ($P$ value is indicated). Source data are available online for this figure.

such a function has not been previously reported for BMAL2. Several studies have indicated that BMAL1 is involved in DNA damage repair (Dakup et al, 2020; Geyfman et al, 2012; Mullenders et al, 2009; Zhang et al, 2023). BMAL1 has some structural and functional similarities to BMAL2. However, the regulation of BMAL2, its interaction with other central clock genes, as well as its downstream targets, are different from those of BMAL1 in normal tissues (Sasaki et al, 2009; Shi et al, 2010). Recent studies have demonstrated that BMAL1 and BMAL2 have distinct, or even opposite, functions in specific tissues and physiological processes (Brady et al, 2016; Mandl et al, 2022; Zhao et al, 2023). Our OCCC experiments found that BMAL2 has an oncogenic effect that is opposite to the previously reported tumor suppressor effect of BMAL1 (Tokunaga et al, 2008). Our data also include the key observation that BMAL2 depletion does not lead to BMAL1 upregulation that could protect cells from DNA damage (Figs. EV7B and EV9A). That data, along with the specificity of GW833972A for BMAL2 and not BMAL1, demonstrate that specific targeting of BMAL2 can be an effective tumor suppression strategy. The low expression of BMAL2 in many non-tumor tissues decreases the risk of negative side effects.

BMAL2 depletion altered the expression of genes in cell cycle, DNA replication, cellular senescence and p53 pathways, in addition to HR, in OCCC cells (Fig. 3A). Thus, targeting BMAL2 can impact multiple tumor-promoting pathways to effectively eliminate OCCC. We demonstrated that the small molecule GW833972A can induce BMAL2 protein degradation. GW833972A monotherapy or combination with PARPi significantly reduced cancer cell viability and impeded tumor growth. As GW833972A has little adverse effects on normal tissues, it could be more beneficial for patients compared to standard chemotherapy with additional drugs, such as PARPi, to achieve synthetic lethality. These findings reveal a new opportunity for OCCC patients, particularly patients with wild-type ARID1A. Lead modification and optimization of GW833972A and detailed pharmacokinetic analyses are worthy of further investigation. Furthermore, such targeting of BMAL2 can be applied more broadly to other BMAL2-upregulated cancers. For example, BMAL2 expression is also associated with poor prognosis in lung cancer, pancreatic cancer, acute myeloid leukemia (AML) and myeloma (Fig. EV9B–E), indicating a pervasive role of BMAL2

in cancer malignancy. Although BMAL2 may carry out its oncogenic functions differently in different tissues, our observations that GW833972A elicits BMAL2 degradation, rather than affecting BMAL2 interaction with one or more other proteins, suggest that GW833972A may be broadly applicable to BMAL2-upregulated cancers even if the mechanism of BMAL2 action differs among those cancers. Both the underlying molecular mechanisms of BMAL2-mediated tumorigenesis and the possibility of using BMAL2 as a multi-cancer therapeutic target are promising areas for further work.

## Methods

**Reagents and tools table**

| Reagent/resource | Reference or source | Identifier or catalog number |
|---|---|---|
| **Experimental models** | | |
| BALB/cAnN.Cg-Foxn1nu/CrlNarl (NUDE) | National Laboratory Animal Center (Taiwan) | RRID:IMSR_CRL:194 |
| ES-2 | Bioresource Collection and Research Center (Taiwan) | 60067 |
| TOV21G | Bioresource Collection and Research Center (Taiwan) | 60407 |
| RMG1 | Japanese Collection of Research Bioresources (Japan) | JCRB0172 |
| OVISE | Japanese Collection of Research Bioresources (Japan) | JCRB1043 |
| KURAMOCHI | Japanese Collection of Research Bioresources (Japan) | JCRB0098 |
| JHOC5 | RIKEN Bioresource Research Center (Japan) | RCB1520 |
| JHOC9 | RIKEN Bioresource Research Center (Japan) | RCB2226 |

| Reagent/resource | Reference or source | Identifier or catalog number |
|---|---|---|
| OVCA429 | Dr. Noriomi Matsumura (Japan) | RRID:CVCL_3936 |
| HeyA8 | Dr. Noriomi Matsumura (Japan) | RRID:CVCL_8878 |
| WI38 | Dr. Liuh-Yow Chen (Taiwan) | RRID:CVCL_0579 |
| HOEC | Applied Biological Materials (Canada) | T4198 |
| HESC | Applied Biological Materials (Canada) | T0533 |
| **Recombinant DNA** | | |
| shBMAL2#1 | RNAiCore (Taiwan) | TRCN21324 |
| shBMAL2#2 | RNAiCore (Taiwan) | TRCN21326 |
| shCtrl (TRC1 Scramble) | RNAiCore (Taiwan) | RNAiCore (Taiwan) |
| pAdenoX-pCMV-I-SceI-mCherry | Dr. Peter Chi | |
| pAdenoX-pCMV-DR-GFP-pPGK-ECFP | Dr. Peter Chi | |
| **Antibodies** | | |
| Anti-BMAL2 | GeneTex | PJ90804 |
| Anti-BMAL2 | Santa Cruz | Sc-365469 |
| Anti-RAD51 | GeneTex | GTX100469 |
| Anti-RAD51 | Cell Signaling Technology | 2577 |
| Anti-γH2AX | Cell Signaling Technology | 9718 |
| Anti-Ki67 | Abclonal | A20018 |
| Anti-p84 | GeneTex | GTX70220 |
| Anti-phospho-histone H3 | Cell Signaling Technology | 3377 |
| Anti-GAPDH | Proteintech | 60004-1-Ig |
| Anti-mouse secondary antibody | Jackson ImmunoResearch | 115-035-003 |
| Anti-rabbit secondary antibody | Jackson ImmunoResearch | 111-035-003 |
| Anti-rabbit CF488A | Sigma-Aldrich | SAB4600036 |
| IgG control | R&D systems | AB-150-C |
| **Oligonucleotides and other sequence-based reagents** | | |
| qPCR primers | This study | See Table EV2 |
| ChIP-qPCR primers | This study | See Table EV3 |
| **Chemicals, enzymes, and other reagents** | | |
| RPMI 1640 medium | Gibco | 31800022 |
| DMEM medium | Gibco | 12100046 |
| McCoy5A medium | Sigma-Aldrich | M4892 |
| DMEM/F12 medium | Gibco | 12500062 |
| MCDB 105 medium | Sigma-Aldrich | M6395 |
| Medium 199 | Sigma-Aldrich | M2520 |
| Ham's F1 medium | Gibco | 21700075 |
| EMEM medium | Sigma-Aldrich | |
| Prigrow X | Applied Biological Materials | TM4198 |

| Reagent/resource | Reference or source | Identifier or catalog number |
|---|---|---|
| Prigrow IV | Applied Biological Materials | TM004 |
| Charcoal-stripped FBS | Gibco | 12676029 |
| L-glutamine | Gibco | 25030081 |
| Fetal bovine serum | Corning | 35-072-CV |
| Trypsin-EDTA (0.25%), phenol red | Gibco | 25200-072 |
| Pen-Strep-Ampho B solution | Biological Industries | 03-033-1B |
| Cycloheximide | Sigma-Aldrich | C1988 |
| DMSO | Sigma-Aldrich | D2650 |
| Olaparib | MedChemExpress | HY-10162 |
| MG132 (Z-Leu-Leu-Leu-al) | MedChemExpress | HY-13259 |
| B02 | Cayman Chemical | 22133 |
| Target Retrieval Solution, Citrate pH 6.1 (10x) | Agilent Dako | S1699 |
| Dako REAL EnVision DAB chromogen | Agilent Dako | K5007 |
| RIPA lysis buffer | Millipore | 20–188 |
| PMSF | Sigma-Aldrich | 11359061001 |
| Protease inhibitors | Sigma-Aldrich | S8830 |
| Phosphatase inhibitors | BIOTOOLS | TAAR-BBI3 |
| Bradford assay (5x) | Bio-Rad | 5000006 |
| Pronase | Merck | 10165921001 |
| TRI Reagent | Merck | T9424 |
| SYBR Green 2X master mix | KAPA Biosystems | KK4619 |
| DAPI | Invitrogen | D1306 |
| Fluorescence mounting medium | Agilent Dako | S3023 |
| Matrigel | Corning | 354234 |
| Benzyl 2-chloro-4 (trifluoromethyl) pyrimidine-5-carboxylate | Matrix Scientific | 003086 |
| 3-Chloroaniline | Thermo Fisher Scientific | 108580050 |
| Methylene chloride | Thermo Fisher Scientific | D151-4 |
| 1,4-dioxane | Thermo Fisher Scientific | 268340010 |
| 4-picolylamine | AK Scientific | J53428 |
| Potassium hydroxide | Showa Chemical | 1637-0150 |
| Other reagents used for the assay | | |
| **Software** | | |
| Prism 10 | GraphPad | RRID:SCR_002798 |
| ImageJ | NIH | RRID:SCR_003070 |
| OpenComet plugin v1.3.1 | OpenComet Software | RRID:SCR_021826 |
| QuPath 0.4.3/0.5.0 | open source | RRID:SCR_018257 |
| Imaris 10.1.0 | Bitplane | RRID:SCR_007370 |
| VisionWorks | Analytik Jena | |
| SynergyFinder 3.0 | FIMM | RRID:SCR_026127 |

| Reagent/resource | Reference or source | Identifier or catalog number |
|---|---|---|
| **Other** | | |
| Mycoplasma Detection Kit | BIOTOOLS | TTB-GBC8 |
| Subcellular Protein Fractionation Kit for Cultured Cells | Thermo Scientific | 78840 |
| ECL detection reagent | Cytiva | RPN2235 |
| CCK8 assay | BIOTOOLS | TEN-CCK8-100 |
| Click-iT EdU Alexa Fluor 488 Imaging Kit | Invitrogen | C10337 |
| Comet assay kit | Abcam | ab238544 |
| Zymo-Spin™ ChIP kit | Zymo Research | D5210 |
| Aperio digital pathology scanner | Leica Biosystems | GT450 |
| Hielscher homogenizer | Hielscher | UP50H |
| UVP ChemStudio Plus Bioimaging | Analytik Jena | N/A |
| Real-Time PCR system | Applied Biosystems | QuantStudio 5 |
| Cell Counter | Olympus | R1 |
| Microplate reader | Tecan | Sunrise |
| Confocal microscope | Leica | Andor Dragonfly 202 |
| Fluorescence microscope | Olympus | |
| Flow Cytometer | BD Biosciences | LSRFortessa |
| Tumor-measuring device | Peira | TM900 |
| Bioruptor Pico Sonicator | Diagenode | N/A |
| High-resolution mass spectrometer | Bruker | MicroTOF-QII |
| HPLC | Agilent | Agilent 1260 Infinity |
| Nanodrop | Thermo | ND 1000 |

## Human tissue samples

Eight OCCC formalin-fixed, paraffin-embedded (FFPE) tissue specimens were purchased from Discovery Life Sciences (USA), and a tissue array containing 6 OCCC cases was purchased from Super Bio Chips (CJ3, Seoul, Korea). Six normal endometrioid epithelium samples were purchased from TissueArray.Com LLC (UT242a, USA). Sample usage was approved by the Institutional Review Board of Human Subjects Research Ethics Committee of Academia Sinica (AS-IRB-BM-23051).

## Cell lines and culture conditions

OCCC cell lines ES-2 (RRID:CVCL_3509) and TOV21G (RRID:CVCL_3613) were purchased from Bioresource Collection and Research Center (Taiwan), OVISE (RRID:CVCL_3116) and RMG1 (RRID:CVCL_1662) and KURAMOCHI (RRID:CVCL_1345) from the Japanese Collection of Research Bioresources (Japan), JHOC5 (RRID:CVCL_4640) and JHOC9 (RRID:CVCL_4643) from the RIKEN Bioresource Research Center (Japan), and OVCA429 (RRID:CVCL_

3936) and HEYA8 (RRID:CVCL_8878) were kindly provided by Dr. Noriomi Matsumura (Kindai University, Japan). Normal fibroblast cell line WI38 (RRID:CVCL_0579) was kindly provided by Dr. Liuh-Yow Chen (Academia Sinica, Taiwan). All cell lines were authenticated using short tandem repeat profiling (Table EV1) and were routinely tested for mycoplasma contamination using a Mycoplasma Detection Kit (BIOTOOLS, TTB-GBC8). OVISE, JHOC9, KURAMOCHI, and HEYA8 were maintained in RPMI supplemented with 10% FBS, OVCA429 in DMEM with 10% FBS, JHOC5 in DMEM/F12 with 10% FBS and 0.1 mM non-essential amino acids, ES-2 in McCoy5a's with 10% FBS, TOV21G in a 1:1 mixture of MCDB 105 and Medium 199 with 15% FBS, RMG1 in Ham's F1 with 10% FBS, WI38 in EMEM with 10% FBS. All culture media contained 1x Pen-Strep-Ampho B solution (Biological Industries, 03-033-1B). Primary human ovarian surface epithelial cells (HOEC, T4198) and human endometrial stromal cells (HESC, T0533) were purchased from Applied Biological Materials (Canada) and maintained on Type I collagen (354249, Corning) coated culture vessels for proper cell adhesion. HOEC was maintained in PriGrow X Series Medium (TM4198, Applied Biological Materials) and HESC in PriGrow IV (TM004, Applied Biological Materials), supplemented with 2 mM L-glutamine (Gibco, 25030081) and 10% charcoal-stripped FBS (Gibco, 12676029). Cells were maintained at 37 °C in a humidified incubator with 5% $CO_2$ and were used within 25 passages after thawing.

## Reagents

The lentiviral shRNAs, shCtrl (TRC1-Scramble_ASN0004), shBMAL2#1 (TRCN21324) and shBMAL2#2 (TRCN21326), were from the National RNAiCore Facility (Academia Sinica). Cycloheximide (CHX; C1988) and DMSO (D2650) were from Sigma-Aldrich. Olaparib (HY-10162) and MG132 (Z-Leu-Leu-Leu-al, HY-13259) were from MedChemExpress. RAD51 inhibitor B02 (22133) was from Cayman Chemical.

## Immunohistochemistry (IHC)

FFPE primary tumor and xenograft sections were used for IHC. Sections were dewaxed with xylene and rehydrated with a descending ethanol series to water. Antigen retrieval was performed using Target Retrieval Solution, Citrate pH 6.1 (10x) (S1699, Agilent Dako) for 10-25 min under high pressure. Endogenous peroxidase was eliminated with 3% $H_2O_2$ for 20 min. Slides were blocked with 5% non-fat milk in PBST for 1 h at room temperature, followed by primary antibodies against BMAL2 (PJ90804, Genetex, 1:2,000), RAD51 (GTX100469, Genetex, RRID:AB_1951602, 1:5,000), γH2AX (9718, Cell signaling technology, RRID:AB_2118009, 1:500), and Ki67 (A20018, Abclonal, RRID:AB_3065688, 1:12,500) diluted in blocking buffer overnight at 4 °C. After washing, slides were incubated with HRP rabbit polymer for 1 h before visualization with liquid diaminobenzidine tetrahydrochloride plus substrate DAB chromogen from Dako REAL EnVision (Dako, K5007, RRID:AB_2888627). All slides were counterstained with hematoxylin. The images were captured under 40X magnification using an Aperio GT450 digital pathology scanner (Leica Biosystems). Positive staining intensity and area were quantified using the QuPath digital analysis system (RRID:SCR_018257).

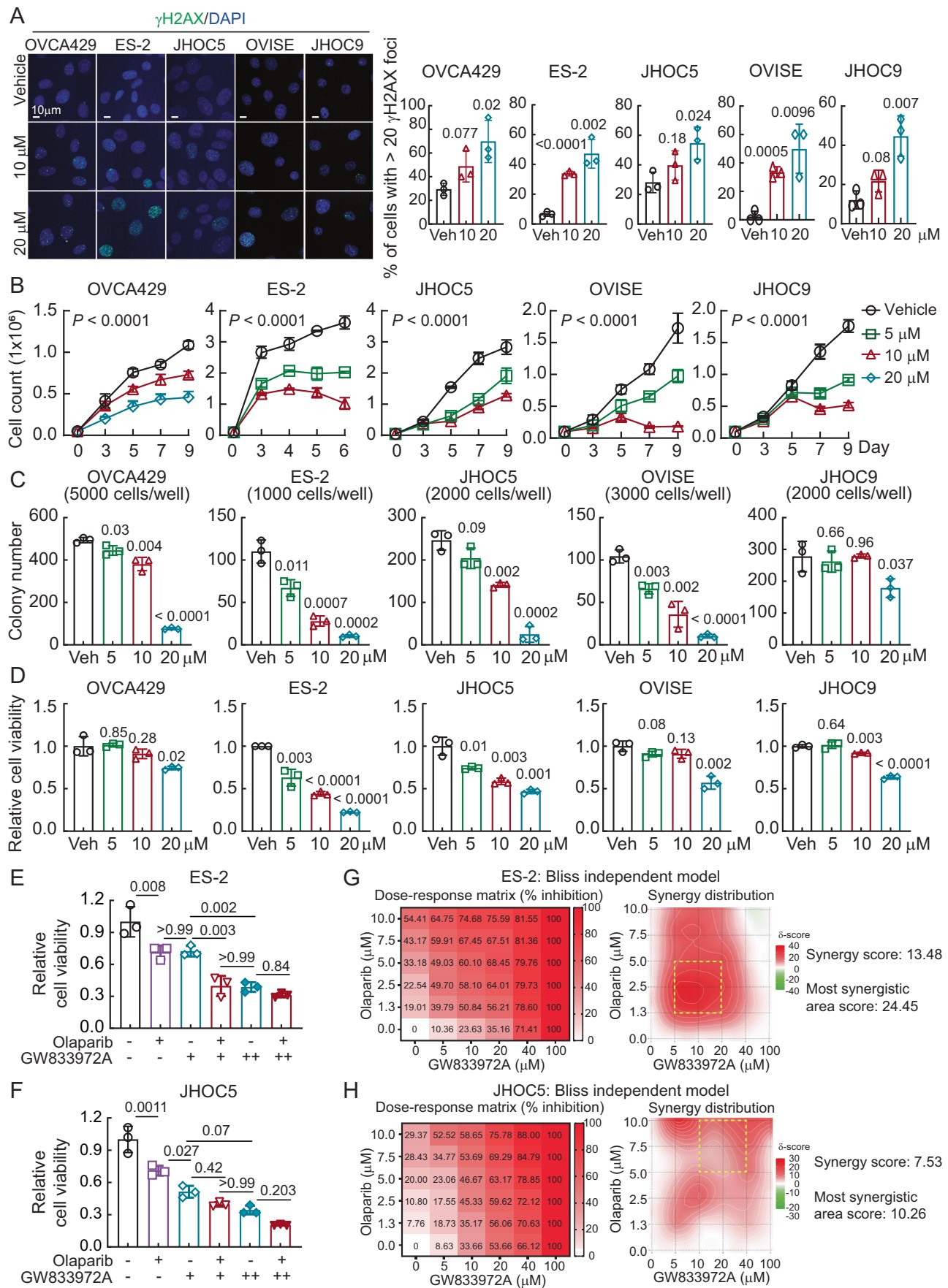

**Figure 5. GW833972A inhibits tumorigenic ability in ARID1A-wt OCCC cells.**

(A) Representative γH2AX immunofluorescent staining of ARID1A-wt OCCC cells treated with vehicle, 10 or 20 μM GW833972A. The percentage of cells with more than 20 γH2AX foci per nucleus was quantified. Data were shown as mean ± SD with $P$ value based on unpaired $t$-test ($n = 3$). Data shown are from one representative experiment of three replicates. (B) Cell growth assays using ARID1A-wt OCCC cells treated with vehicle, 5, 10, or 20 μM GW833972A. Data were shown as mean ± SD with $P$ value based on two-way ANOVA test ($n = 3$). Data shown are from one representative experiment of three replicates. (C) Clonogenic assays using ARID1A-wt OCCC cells treated with vehicle, 5, 10, or 20 μM GW833972A. Data were shown as mean ± SD with $P$ value based on unpaired $t$-test ($n = 3$). Data shown are from one representative experiment of three replicates. (D) Cell viability assays using ARID1A-wt OCCC cells treated with vehicle, 5, 10, or 20 μM GW833972A. Data were shown as mean ± SD with $P$ value based on unpaired $t$-test ($n = 3$). Data shown are from one representative experiment of three replicates. (E) Cell viability assays using ES-2 cells treated with vehicle, 10 (+) or 20 (++) μM GW833972A, with vehicle, 5 (+) or 10 (++) μM Olaparib. Data were shown as mean ± SD with $P$ value based on one-way ANOVA test ($n = 3$). Data shown are from one representative experiment of three replicates. (F) Cell viability assays using JHOC5 cells treated with vehicle, 10 (+) or 20 (++) μM GW833972A, with vehicle, 2.5 (+) or 5 (++) μM Olaparib. Data formatting is as described for (E). (G, H) Combination effect of GW833972A and Olaparib in ES-2 (G) and JHOC5 (H) cells using the Bliss independence model. The synergy scores were calculated using SynergyFinder 3.0. Yellow boxes indicate the most synergistic area. Source data are available online for this figure.

## Immunoblot (IB) assay

Whole cell lysate was prepared using RIPA lysis buffer (Millipore, 20–188) with 1 mM PMSF (11359061001, Sigma-Aldrich), protease inhibitors (Sigma-Aldrich, S8830) and phosphatase inhibitor cocktail (BIOTOOLS, TAAR-BBI3) followed by sonication at 4 °C using a UP50H (Hielscher). Nuclear proteins were extracted using the Subculture Protein Fractionation Kit for Cultured Cells (78840, Thermo Scientific) according to the manufacturer's instructions to detect γH2AX level. Protein concentration was determined by Bradford assay (5000006, Bio-Rad). Immunoblotting (IB) was performed after SDS-PAGE, with overnight incubation with primary antibodies against BMAL2 (sc-365469, Santa Cruz Biotechnology, 1:500), BMAL1 (sc365645, Santa Cruz, RRID:AB_10841724, 1:1000), RAD51 (8875, Cell signaling technology, RRID:AB_2721109, 1:1000) or γH2AX (2577, Cell signaling technology, RRID:AB_2118010, 1:1,000) followed by 1:10,000 dilution of horseradish peroxidase-conjugated anti-mouse antibody (115-035-003; Jackson ImmunoResearch, RRID:AB_10015289, 1:10,000) or anti-rabbit (111-035-003, Jackson ImmunoResearch, RRID: AB_2313567, 1:10,000). p84 (GTX70220, GeneTex, RRID: AB_372637, 1:2000) and GAPDH (60004-1-Ig, Proteintech, RRID:AB_2107436, 1:10,000) were used as loading controls for nuclear protein and whole cell protein, respectively. Signals were detected using Amersham ECL Select Western Blotting Detection Reagent (RPN2235, Cytiva) and images captured by a UVP ChemStudio Plus BioImaging system (Analytik Jena). The intensity of the bands was quantified using ImageJ software (RRID:SCR_003070).

## Cellular thermal shift assay (CETSA)

Cells were treated with 20 μM GW833972A or vehicle (DMSO) for 1 h at 37 °C before being subjected to 3-min heat treatment at the indicated temperature (25–60 °C). For each temperature condition, $5 \times 10^5$ cells in 100 uL PBS were used. After heat treatment, the cells were immediately placed on ice for 5 min and then lysed in 0.1% NP-40 lysis buffer (50 mM HEPES, pH 7.5, 100 mM NaCl, 0.1% Igepal, and 1 mM EDTA) containing protease inhibitors by gentle pipetting. The lysates were centrifuged at $20,000 \times g$ for 20 min at 4 °C to remove the insoluble aggregates. The soluble fractions were used to determine BMAL2 thermal stability using IB assay after SDS-PAGE. The intensity of the bands was quantified using ImageJ software.

## Drug affinity responsive target stability (DARTS) assay

Cell lysates were prepared in 0.1% NP-40 lysis buffer and treated with vehicle (DMSO), 1 or 10 μM GW833972A for 1 h at room temperature with gentle end-to-end rotation, followed by 0.2 μg/mL pronase (#10165921001, Merck) treatment for 10 min to induce limited proteolysis. The proteolysis was stopped by the addition of SDS sample buffer with 1x protease inhibitor and heated at 100 °C for 5 min. The BMAL2 protein level was determined by IB assay after SDS-PAGE. The intensity of the bands was quantified using ImageJ software.

## RNA extraction and qRT-PCR

Total RNA was extracted using TRI Reagent (T9424, Merck), treated with DNase I (EN0521, Thermo Fisher Scientific) and cDNA reverse transcribed with Invitrogen SuperScript III Reverse Transcriptase (18-080-044, Invitrogen). qRT-PCR was performed using Applied Biosystems QuantStudio 5 Real-Time PCR System with SYBR GREEN 2X master mix (KK4619, KAPA Biosystems). The mRNA relative quantities were determined using comparative cycle threshold methods with RNA18S5 as an internal control. Primer sequences are listed in Table EV2.

## Cell growth, clonogenic, and cell viability assays

To determine growth curves, $1 \times 10^5$ cells were seeded in each well of a six-well plate and cultured under the indicated conditions. Cell numbers were counted at four time points using Olympus R1 Cell counter. Unpaired $t$-test was used to test for significant differences between shCtrl and shBMAL2 or vehicle and treatment groups. To evaluate cell survival ability, a clonogenic assay was performed using $1 \times 10^3$ cells seeded into 6- or 12-well plates and cultured for at least 7 days. Colonies were fixed and stained with 0.5% (w/v) crystal violet in methanol. Clonogenicity was either measured by colony numbers quantified using VisionWorks software (Analytik Jena), or by extracting the colonies with 15% methanol and 15% acetic acid and measuring the absorbance of the extracted dye at 570 nm. For cell viability assays, 1000 or 2000 cells were seeded in each well of 96-well plates and cultured under the indicated conditions. Cell viability was analyzed using Cell Counting Kit-8 (CCK8; TEN-CCK8-100, BIOTOOLS) following the manufacturer's instructions. Absorbance was measured at 450 nm using a

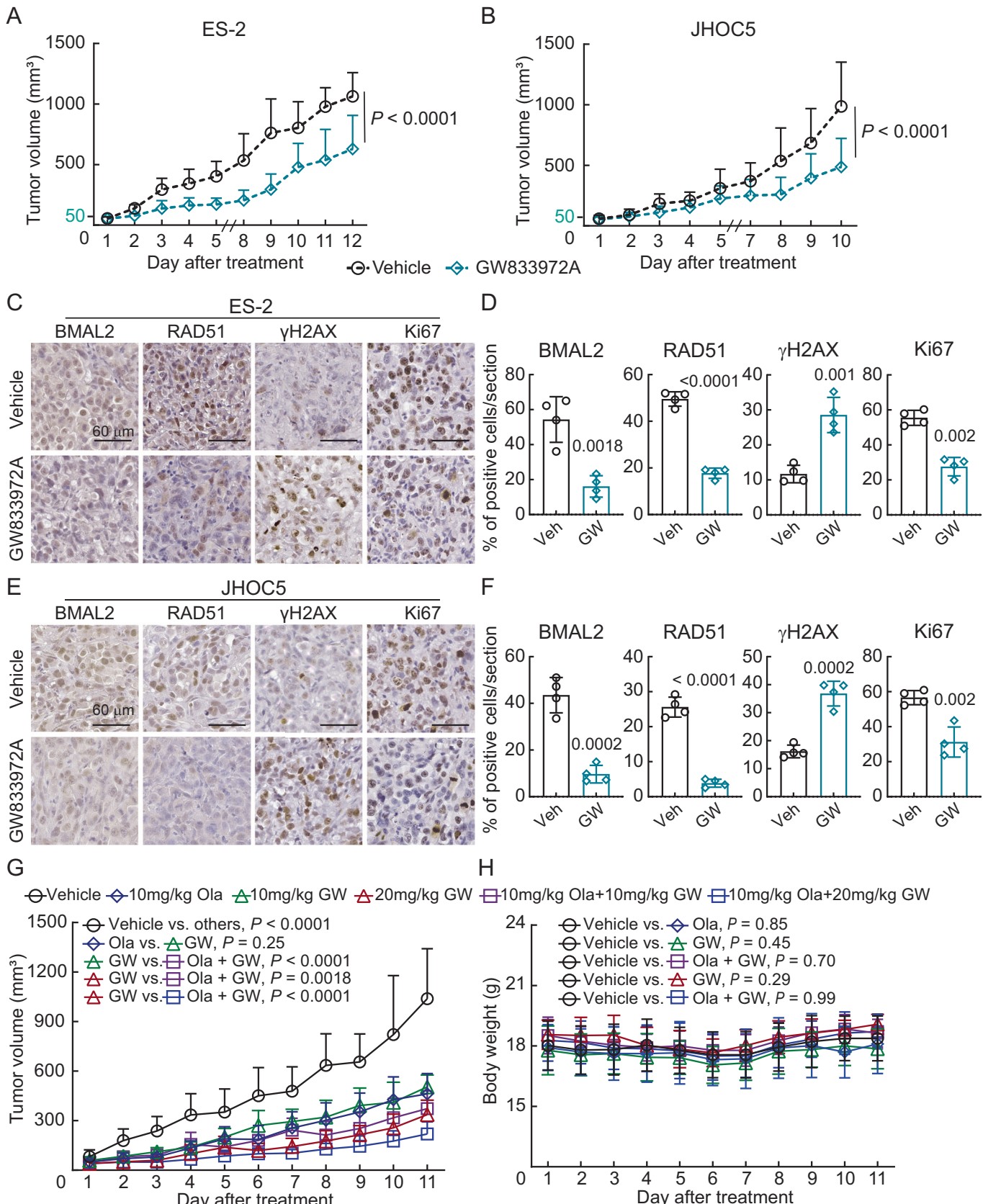

◀

**Figure 6. Targeting BMAL2 by GW833972A increases DNA damage and inhibits ARID1A-wt OCCC tumor growth in mouse xenografts.**

(A) Subcutaneous xenograft model in NUDE mice using ES-2 cells with or without 10 mg/kg GW833972A treatment. Five mice were used for each group, and data were shown as mean ± SD ($n = 5$). Nonlinear regression (curve fit) analysis was used to test for significant differences between the tumor growth curve of the vehicle control and GW833972A-treated group ($P$ value is indicated). (B) Subcutaneous xenograft model in NUDE mice using JHOC5 cells with or without 10 mg/kg GW833972A treatment. Five mice were used for each group, and data were shown as mean ± SD ($n = 5$). Data formatting is as described for (A). (C) Representative images of BMAL2, RAD51, γH2AX, and Ki67 IHC staining using serial tumor sections from ES-2-derived tumors. (Scale bars, 60 μm). (D) Quantification of the percentage of positive cells for the experiment shown in (C). Randomly selected regions from tissue sections shown in (C) were used ($n = 4$) for staining and quantification. Data were shown as mean ± SD with $P$ value based on an unpaired $t$-test. (E) Representative images of BMAL2, RAD51, γH2AX and Ki67 IHC staining using serial tumor sections from JHOC5-derived tumors. (Scale bars, 60 μm). (F) Quantification of the percentage of positive cells for the experiment shown in (E). Randomly selected regions from tissue sections shown in (E) were used ($n = 4$) for staining and quantification. Data were shown as mean ± SD with $P$ value based on an unpaired $t$-test. (G) Subcutaneous xenograft model in NUDE mice using ES-2 cells treated with either vehicle, 10 mg/kg Olaparib, 10 mg/kg or 20 mg/kg GW833972A, or GW833972A-Olaparib combination. Five mice were used for each group, and data were shown as mean ± SD ($n = 5$). Nonlinear regression (curve fit) analysis was used to test for significant differences between the tumor growth curves of different groups ($P$ values are indicated). (H) Body weight curve of the mice from (G). Nonlinear regression (curve fit) analysis was used to test for significant differences between the body weight curve of the vehicle control and drug-treated groups ($P$ values are indicated). Source data are available online for this figure.

microplate reader (Sunrise, TECAN). Unpaired $t$-test was used to compare shCtrl and shBMAL2 or vehicle and treatment groups.

## Immunofluorescence (IF) staining

Cells were seeded and left to adhere to reach 70–80% confluence on coverslips and then fixed with 4% paraformaldehyde for 20 min at room temperature, followed by permeabilization with 0.5% Triton X-100 for 20 min. The cells were then blocked with 1% bovine serum albumin (BSA)/PBS for 1 h at room temperature and incubated with primary antibodies against phospho-histone H3 (pHH3; #3377 s, Cell signaling technology; RRID:AB_1549592, 1:800) or γH2AX (#2577 s, Cell signaling technology, 1:400) in 1% BSA/PBS overnight at 4 °C. After washing with PBS, the cells were incubated with anti-rabbit CF488A (SAB4600036, Sigma-Aldrich; RRID:AB_2728116, 1:300) secondary antibody for 1 h at room temperature in the dark. 4'6-diamidino-2-phenylindole (DAPI, D1306, Invitrogen; RRID:AB_2629482) was used to stain nucleus. Coverslips with stained cells were mounted onto glass slides with fluorescence mounting medium (S3023, Agilent Dako). Images were acquired using an Andor Dragonfly 202 high-speed confocal microscope system, and the level of fluorescence signal was measured by fluorescence emission at 505-550 nm with excitation at 488 nm. The fluorescence intensity was quantified using the Imaris 10.1.0 software (RRID:SCR_007370).

Click-iT EdU Alexa Fluor 488 Imagin Kit (C10337, Invitrogen) was used to evaluate cell proliferation according to the manufacturer's protocol. Nuclei were counterstained with Hoechst before mounting. Images were acquired using a fluorescence microscope (Olympus), and the percentage of EdU-positive cells was quantified using VisionWorks (Analytik Jena).

## Adenovirus-based HR activity reporter assay

Adenovirus-based HR activity reporter assay was performed as previously described (Lee et al, 2023). Cells were transduced with pAdenoX-pCMV-I-SceI-mCherry alone, pAdenoX-pCMV-DR-GFP-pPGK-ECFP alone or both recombinant adenoviruses, then harvested and analyzed using a LSRFortessa flow cytometer (BD Biosciences) three to four days later. The proportion of EGFP-positive cells was determined to represent HR activity.

## Subcutaneous xenograft model

Animal care and experiments were approved by the Institutional Animal Care and Utilization Committee of Academia Sinica (IACUC# 21-12-1760). Female BALB/cAnN.Cg-Foxn1nu/CrlNarl (NUDE) (RRID:IMSR_CRL:194) mice were purchased from the National Laboratory Animal Center (Taipei, Taiwan) at 6–8 weeks of age and housed in a P1-level (ABSL-1) animal room that meets the SPF (specific-pathogen-free) standards. Subcutaneous tumor xenograft models were performed with injection of $2 \times 10^6$ ES-2, $5 \times 10^6$ JHOC5, or $5 \times 10^6$ OVISE cells mixed with Matrigel (354234, Corning) at a 1:1 (v/v) ratio into the lower flanks of each mouse. Tumor size was measured every other day using a Peira TM900 tumor volume-measuring device. The experiment was terminated 4 weeks after injection or when tumors reached a 2-cm diameter. For drug treatment experiments, mice were randomly assigned to different groups when the subcutaneous tumors reached 50 mm³. Tumor growth was monitored every day by Peira TM900. Mice were euthanized when tumors in the control group reached a 2-cm diameter or when the tumor-induced skin breakage occurred in the vehicle control group.

## Comet assay

Comet Assay Kit (ab238544, Abcam) was used to evaluate DNA damage. Briefly, cells were embedded in low-melting-point agarose on comet slides and lysed to remove membranes and soluble cellular components. Slides were subjected to alkaline electrophoresis to allow DNA migration. The slides were stained with Vista Green Dye, and images were captured using a fluorescence microscope (Olympus). DNA damage was quantified by measuring tail moment using the OpenComet plugin (RRID:SCR_021826) in ImageJ software. At least 50 cells per sample were analyzed in each condition.

## Chromatin immunoprecipitation (ChIP)

ChIP assays were performed using the Zymo-Spin™ ChIP kit (D5210, Zymo Research) following the manufacturer's instructions. Briefly, cells were crosslinked with 1% formaldehyde for 10 min at room temperature, and the crosslinking was stopped by 0.125 M glycine. Cross-linked chromatin was sheared using a Bioruptor Pico sonicator (Diagenode) to a size with a range between 100 and 110 bases. Immunoprecipitations were performed with antibodies against BMAL2 (PJ90804, GeneTex) or IgG control (AB-150-C,

R&D Systems) and ZymoMag Protein A (M2001) magnetic beads. qPCR was performed to detect protein-associated promoter regions using primers listed in Table EV3. Data were normalized to input and presented as fold enrichment relative to IgG control.

## Structure-based virtual screening of BMAL2 inhibitor

The structure-based virtual screening was performed by TargetMol Chemicals Inc. (USA). The C-terminal PAS2 domain (N360-E477) of BMAL2, resolved by the NMR technique [Protein Database (PDB) Entry- 2KDK] and the Alphafold protein structure database-predicted BMAL2 full-length 3D structure were used to screen for potential binding pockets. The small molecule binding pocket was detected by using MOE-SiteFinder, where the pocket served as the docking area for the virtual screening. Dataset T001, including 18,100 bioactive compounds and natural products, and dataset LF1000, containing 50000 purchasable compounds, were used to screen potential modulators of BMAL2. The program OEDock (Ver 3.2.0.2) (RRID:SCR_002970) was used for virtual screening, and the compounds with high BMAL2 binding affinity were obtained by the Chemgauss4 scoring function. Based on the docking score, predicted drug-like properties, and binding mode analysis, 119 high-affinity compounds (FRED Chemgauss4 score between −9.3 and −7.8) were identified. The top nine high-affinity compounds with favorable drug properties were selected. Candidate compounds were tested at 5 and 10 μM in cell-based assays to identify BMAL2 inhibitors.

## GW833972A synthesis

GW833972A was synthesized in-house as outlined in the schematic diagram below. The compound was prepared through a three-step synthetic route starting from the commercially available benzyl 2-chloro-4-(trifluoromethyl)pyrimidine-5-carboxylate (**1**). Reaction of (**1**) with 3-chloroaniline in 1,4-dioxane at room temperature via a nucleophilic aromatic substitution (SNAr) afforded benzyl 2-((3-chlorophenyl)amino)-4-(trifluoromethyl)pyrimidine-5-carboxylate (**2**). Subsequent base-promoted hydrolysis of (**2**) yielded 2-((3-chlorophenyl)amino)-4-(trifluoromethyl)pyrimidine-5-carboxylic acid (**3**), which was then coupled with 4-picolylamine in the presence of EDC·HCl and HOBt·H₂O to furnish GW833972A. All reagents, solvents, and starting materials were commercially purchased. Benzyl 2-chloro-4 (trifluoromethyl)pyrimidine-5-carboxylate was from Matrix Scientific. 3-Chloroaniline, methylene chloride and 1,4-dioxane were from Thermo Fisher Scientific. 4-picolylamine was from AK Scientific, ethanol (EtOH) was from Shimakyu Chemical, and potassium hydroxide was from Showa Chemical. The reactions were monitored by thin-layer chromatography (TLC) plates using Merck silica gel 60 F254. A handheld UV lamp manufactured by Analytik Jena, UVG-11 (254 and 365 nm), was used to identify the spots. All synthesized compounds were characterized by 1H, 13C, and 19F NMR spectra recorded on a Bruker AVIII HD 400 MHz spectrometer. The chemical shifts are reported in ppm and were internally referenced to the residual solvent signals: for CDCl₃, δ 7.26 ppm for ¹H NMR and δ 77.16 ppm for ¹³C NMR; for DMSO-d₆, δ 2.50 ppm for ¹H NMR and δ 39.52 ppm for ¹³C NMR, respectively. The coupling constants (J) were reported in Hz, and the splitting patterns were singlet, doublet, triplet, quartet, multiplet, and broad peaks, abbreviated as s, d, t, q, m, and bs, respectively. High-

resolution mass spectroscopy (HRMS) was obtained on a Bruker microTOF-QII mass spectrometer connected with an Agilent 1260 Infinity HPLC. The ionization source parameters were ESI positive, nebulizer (2.5 Bar), capillary (4.5 kV), dry heater (220 °C), end plate offset (−500 V), scan region (50–3000 m/z), dry gas (8.0 L/min), and collision cell RF (250.0 Vpp). Detailed structural characterization data can be found in Appendix.

## RNA sequencing (RNA-seq) analysis

Total RNA was extracted and quantified using NanoDrop (Thermo, ND 1000). All samples with an OD 260/280 ratio >1.8 and an OD 230/260 ratio >2 were sent to BIOTOOLS Company in Taiwan for sequencing. RNA quality was measured by the Qsep100 Bio-Fragment Analyzer. Samples with RQN >6.8 were used for cDNA library construction. Paired-end 150-bp reads, sequenced by the NovaSeq 6000 platform, were first processed using Trimmomatic (v0.38) (RRID:SCR_011848) to trim adapters and filter out low-quality bases. The trimmed reads were then aligned to the human reference genome (GRCh38) using HISAT2 (v2.1.0) (RRID:SCR_015530). The Gene-level read counts were subsequently quantified using featureCounts (v2.0.0) (RRID:SCR_012919). After normalization of the count data, differentially expressed genes (DEGs) between shBMAL2 and shCtrl for each cell line were determined based on an adjusted $p$ value <0.05 and an absolute log2 fold change >1 using the R package "DESeq2" (v1.26.0) (RRID:SCR_015687). To elucidate the biological pathways associated with the identified DEGs, Kyoto Encyclopedia of Genes and Genomes (KEGG) (RRID:SCR_012773) enrichment analysis was conducted using the R package "clusterProfiler" (v3.14.3) (RRID:SCR_016884). For each cell line, the expression profile of each shBMAL2 clone was compared with the shCtrl and the expression fold change of genes in the KEGG-HR pathway (hsa03440) was calculated (Dataset EV1).

## Statistics

For cell-based assays, all experiments were performed in at least three independent biological replicates, with each replicate consisting of technical triplicates unless otherwise noted. For xenograft models, a power analysis was used to determine the minimum sample size required to detect a statistically significant effect (α = 0.05, 1 − β = 0.8) and also to adhere to the 3Rs principle. Only female NUDE mice at 6–8 weeks of age were used. The mice were randomly assigned to the shCtrl or shBMAL2 group for subcutaneous injection. For drug treatment experiments, mice were randomly assigned to different groups when the subcutaneous tumors reached 50 mm³. For each experiment, all data points were included for $t$-test, one-way ANOVA followed by Tukey's multiple comparisons test, or nonlinear regression (curve fit) analysis to compare control and treatment groups. No blinding was done. Kaplan–Meier estimation method was used for overall survival and progression-free survival analyses, and a log-rank test was used to compare differences. Statistical analyses were performed using Prism 10 (RRID:SCR_002798).

# Data availability

RNA-Seq data: European Nucleotide Archive (ENA) PRJEB101975.

The source data of this paper are collected in the following database record: biostudies:S-SCDT-10_1038-S44321-026-00414-8.

## The paper explained

### Problem

Ovarian clear cell carcinoma (OCCC) is highly chemoresistant and has worse clinical outcomes at advanced stages than other ovarian cancer subtypes. The survival rate for stage III and IV OCCC patients is ~30 and 0%, respectively. Mutation of the AT-rich interactive domain 1A gene (ARID1A) which leads to ARID1A deficiency is frequently found in OCCC. However, a substantial portion of OCCC cases retain ARID1A expression. These OCCC cases are biologically different from ARID1A-deficient OCCC. Particularly, targeted therapies that sensitize ARID1A-deficient OCCC to chemotherapy in preclinical settings are largely ineffective against OCCC with wild-type (wt) ARID1A. Thus, it is important to identify druggable targets and develop targeted therapies for ARID1A-wt OCCC and other cancers with similar mechanisms of chemoresistance.

### Results

We found that BMAL2 was upregulated in both ARID1A-wt and ARID1A-mut OCCC cells. Depleting BMAL2 led to DNA damage accumulation and reduced tumor growth by inhibiting DNA repair pathways. This indicated that dependence on high BMAL2 expression is a vulnerability that could be used to inhibit the progression of OCCC. Consistent with this hypothesis, we also discovered that a small molecule, GW833972A, binds to and facilitates BMAL2 degradation, and thereby inhibits tumor growth with little adverse side effects in xenograft models.

### Impact

BMAL2 depletion, or degradation by a small molecule, led to DNA damage accumulation, decreased cell viability and reduced tumorigenesis of ARID1A-wt OCCC, indicating that BMAL2 is an appealing therapeutic target. As GW833972A promoted BMAL2 degradation, rather than inhibiting specific protein interactions, it may be broadly applicable to cancers with upregulated BMAL2.

## Peer review information

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

## Acknowledgements

This work was supported by Academia Sinica (AS) [AS-GCS-111-L01] and the Taiwan National Science and Technology Council [NSTC-114-2320-B-001-016-MY3] grants to Wendy W. Hwang-Verslues. The authors would like to thank Dr. Pang-Hung Hsu (Department of Bioscience and Biotechnology, National Taiwan Ocean University) for thorough discussion about the structure-based virtual screening results, Dr. Chao-Chin Li, a veterinarian (Institute of Cellular and Organismal Biology, Academia Sinica), for confirming the IHC results and the following core facilities at Academia Sinica: the National RNAiCore Facility for providing shRNA reagents and related services; the Bioinformatics Core at Institute of Molecular Biology for providing the RNA-seq analysis services; the DNA Sequencing Core Facility of the Institute of Biomedical Sciences [funded by Academia Sinica Core Facility and Innovative Instrument Project (AS-CFII-113-A12)] for DNA sequencing analysis; the Advanced Optics Microscope Core Facility (funded by AS-CFII-114-A3) for microscope imaging technical support, and Academia Sinica SPF Animal Facility (AS-CFII-113-A7) for providing animal support.

## Author contributions

**Grace Y T Tan**: Conceptualization; Formal analysis; Validation; Investigation; Visualization. **Pei-Yi Lin**: Formal analysis; Investigation; Visualization. **Li-Tzu Cheng**: Investigation. **Yu-Sheng Tsai Yuan**: Investigation. **Shih-Han Huang**: Investigation. **Chen-Hsin Albert Yu**: Formal analysis. **Chao-Tsen Chen**: Resources. **Peter Chi**: Resources. **Wendy W Hwang-Verslues**: Conceptualization; Formal analysis; Supervision; Funding acquisition; Validation; Visualization; Writing—original draft; Project administration; Writing—review and editing.

Source data underlying figure panels in this paper may have individual authorship assigned. Where available, figure panel/source data authorship is listed in the following database record: biostudies:S-SCDT-10_1038-S44321-026-00414-8.

## Disclosure and competing interests statement

The authors declare no competing interests.

# Expanded View Figures

**Figure EV1.  BMAL2 depletion inhibits tumorigenic ability in OCCC cells.**

(**A**) Representative EdU staining of OCCC cells without (shCtrl) or with BMAL2 depletion (shBMAL2#1 or #2). Scale bar indicates 100 μm. (**B**) Representative phospho-histone H3 (pHH3) staining of shCtrl and shBMAL2 OCCC cells. Scale bar indicates 200 μm. (**C**) Representative images of clonogenic assays in OCCC cells. (**D**) Individual 3D ultrasound tumor images from xenograft models in NUDE mice using shCtrl or shBMAL2 ES-2, JHOC5 and OVISE cells. Five mice were used for each group. The size of each tumor is indicated. (**E**) Representative images of BMAL2 and Ki67 IHC staining using serial tumor sections from ES-2, JHOC5 or OVISE derived tumors. Scale bars indicate 60 μm. Source data are available online for this figure.

▶

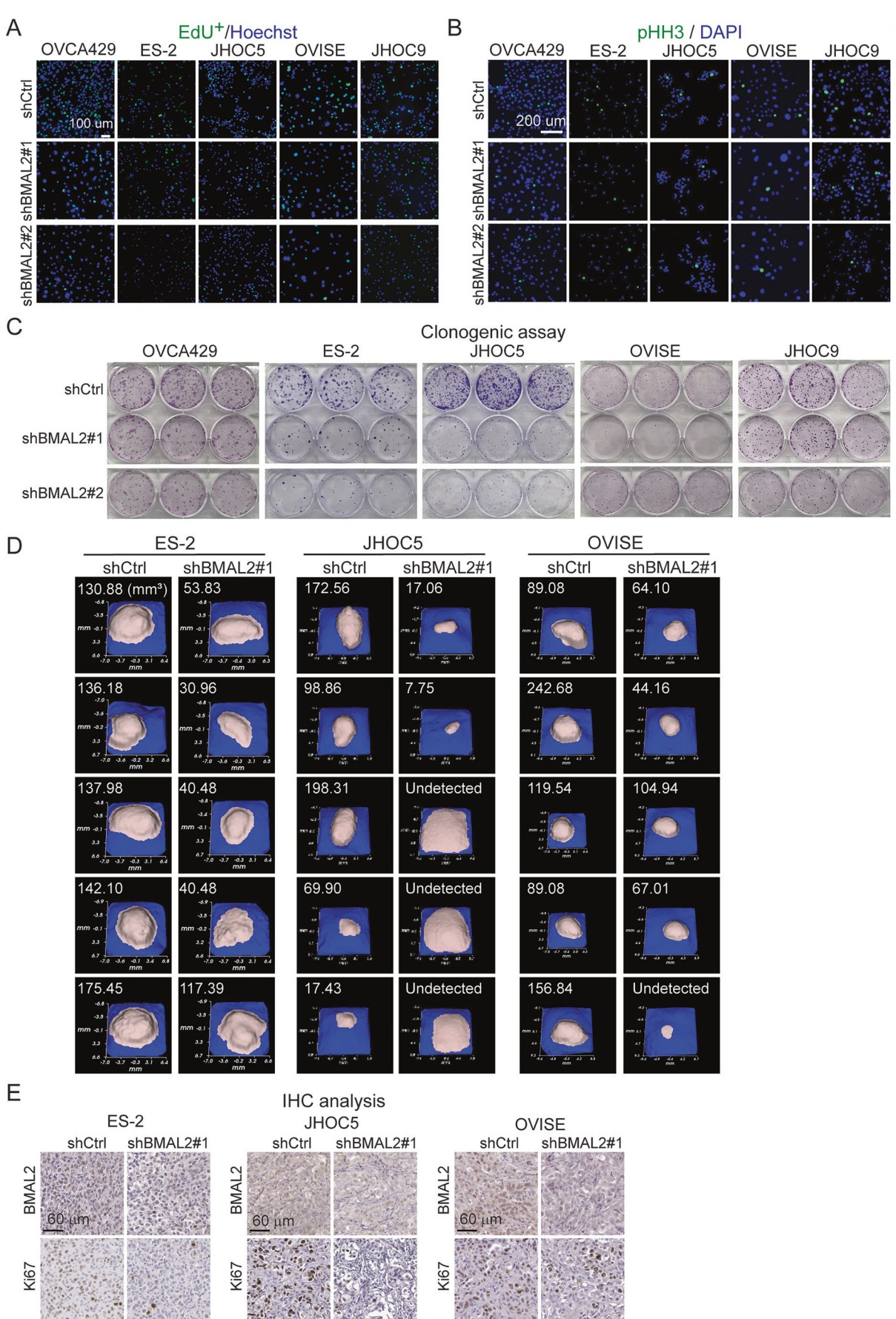

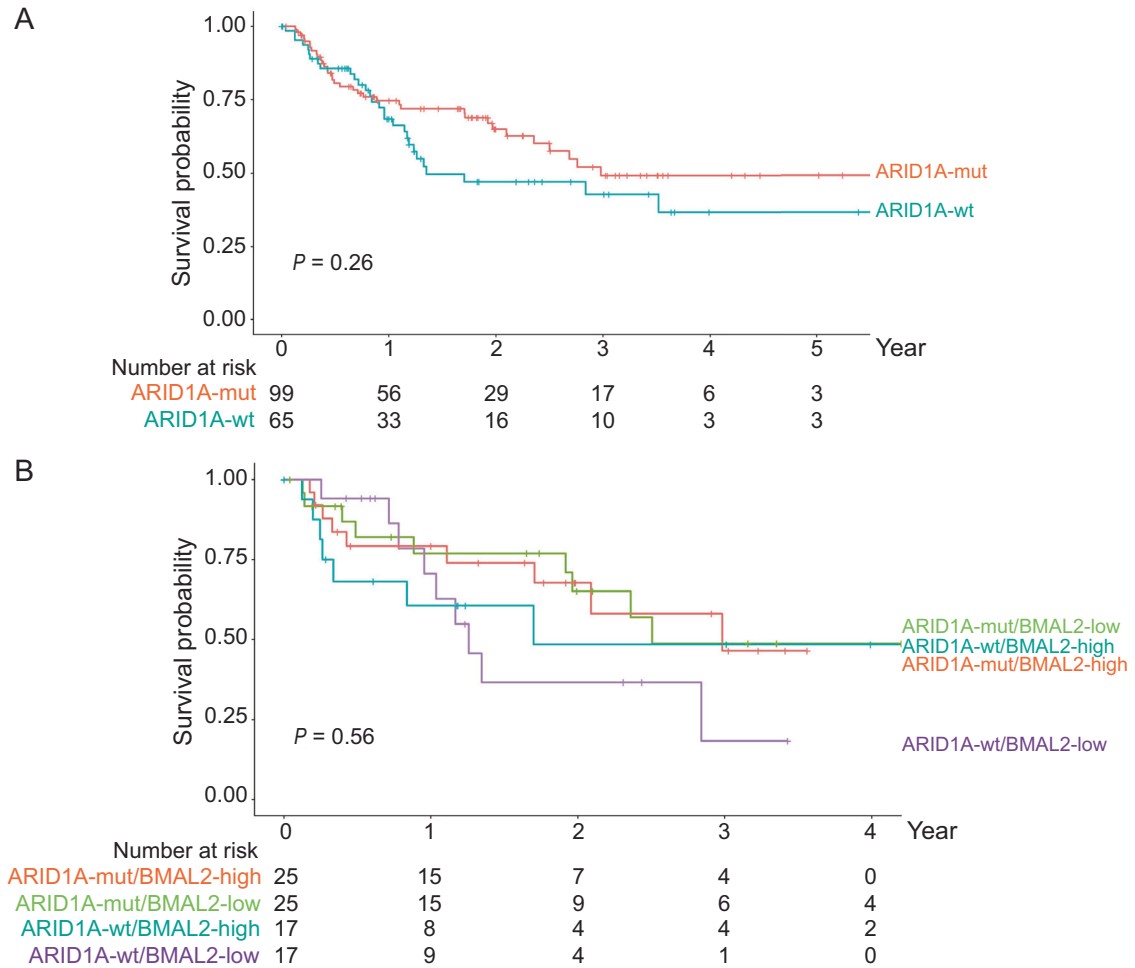

**Figure EV2. BMAL2 function may be independent of ARID1A status in OCCC.**

(A) Kaplan–Meier overall survival (OS) analysis of OCCC patients grouped by ARID1A status. ARID1A-mut group ($n = 99$) is indicated by the red line; ARID1A-wt group ($n = 65$) is indicated by the cyan line. $P = 0.26$. The $P$ value was determined by the log-rank test. (B) Kaplan–Meier OS analysis of ARID1A-mut and ARID1A-wt OCCC patients grouped by BMAL2 expression level. The first (BMAL2-high) and fourth (BMAL2-low) quartiles, both within and between ARID1A-wt ($n = 17$ for each group) and ARID1A-mut ($n = 25$ for each group) cases, were compared. ARID1A-mut/BMAL2-high group ($n = 25$) is indicated by red line; ARID1A-mut/BMAL2-low group ($n = 25$) is indicated by green line; ARID1A-wt/BMAL2-high group ($n = 17$) is indicated by cyan line; ARID1A-wt/BMAL2-low group ($n = 17$) is indicated by purple line. $P = 0.56$. The $P$ value was determined by the log-rank test.

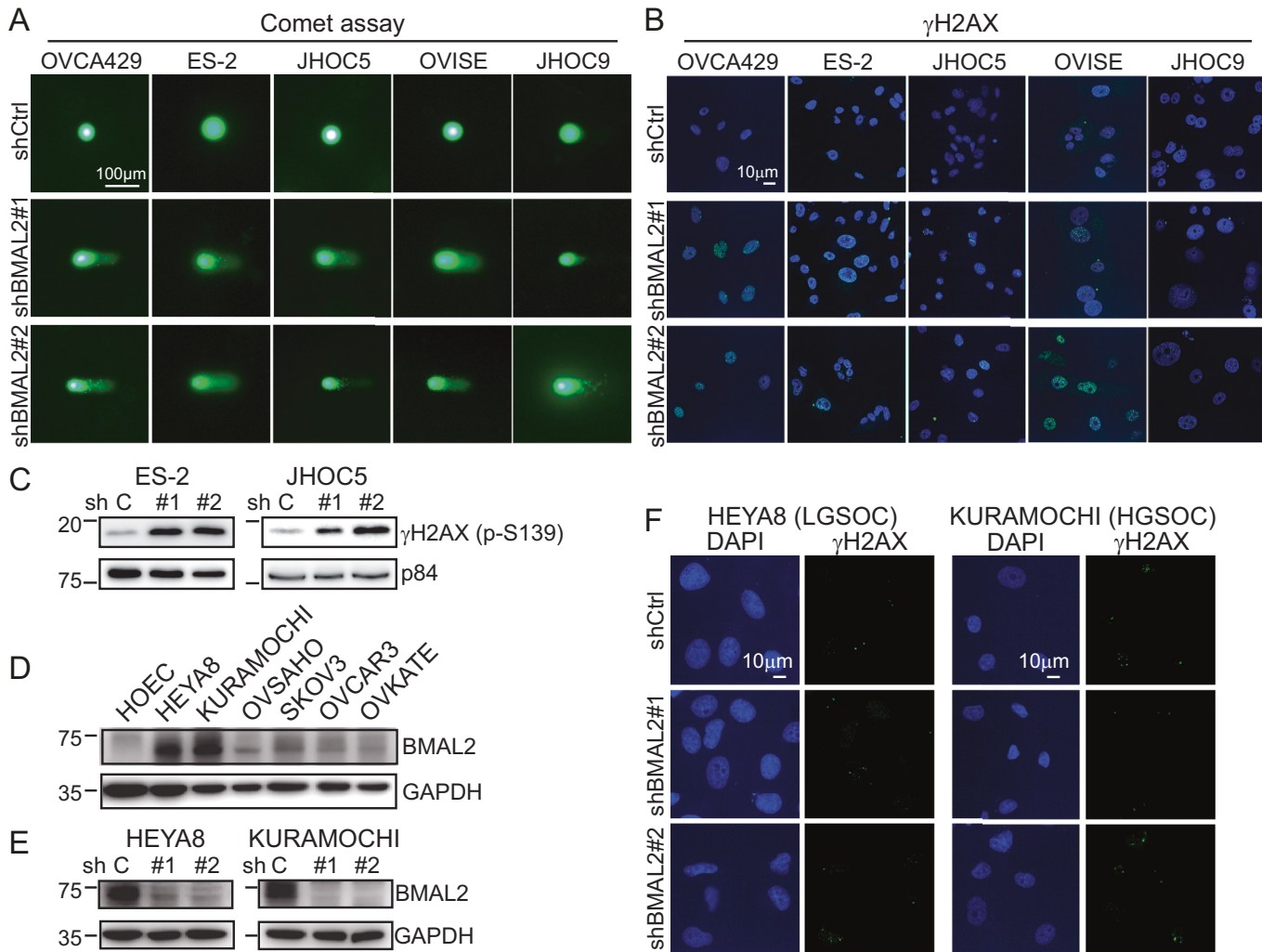

**Figure EV3. BMAL2 depletion increases endogenous DNA damage in OCCC cells, but not serous type ovarian cancers.**

(A) Representative comet assay images of OCCC cells without (shCtrl) or with BMAL2 depletion (shBMAL2#1 or #2). These cells were not treated with DNA damage agents. Scale bar indicates 100 μm. (B) Representative γH2AX staining of shCtrl and shBMAL2 OCCC cells. These cells were not treated with DNA damage agents. Scale bar indicates 10 μm. (C) IB of γH2AX protein with p84 as a nuclear protein loading control. Blots shown are from one representative experiment of three replicates. (D) IB of BMAL2 protein expression with GAPDH as a loading control in serous ovarian cancer cell lines. Blots shown are from one representative experiment of three replicates. (E) IB of BMAL2 protein expression with GAPDH as a loading control in BMAL2-depleted (shBMAL2#1 or #2) HEYA8 and KURAMOCHI cells. Blots shown are from one representative experiment of three replicates. (F) Representative γH2AX staining of shCtrl and shBMAL2 serous type OC cells. These cells were not treated with DNA damage agents. Scale bars indicate 10 μm. Source data are available online for this figure.

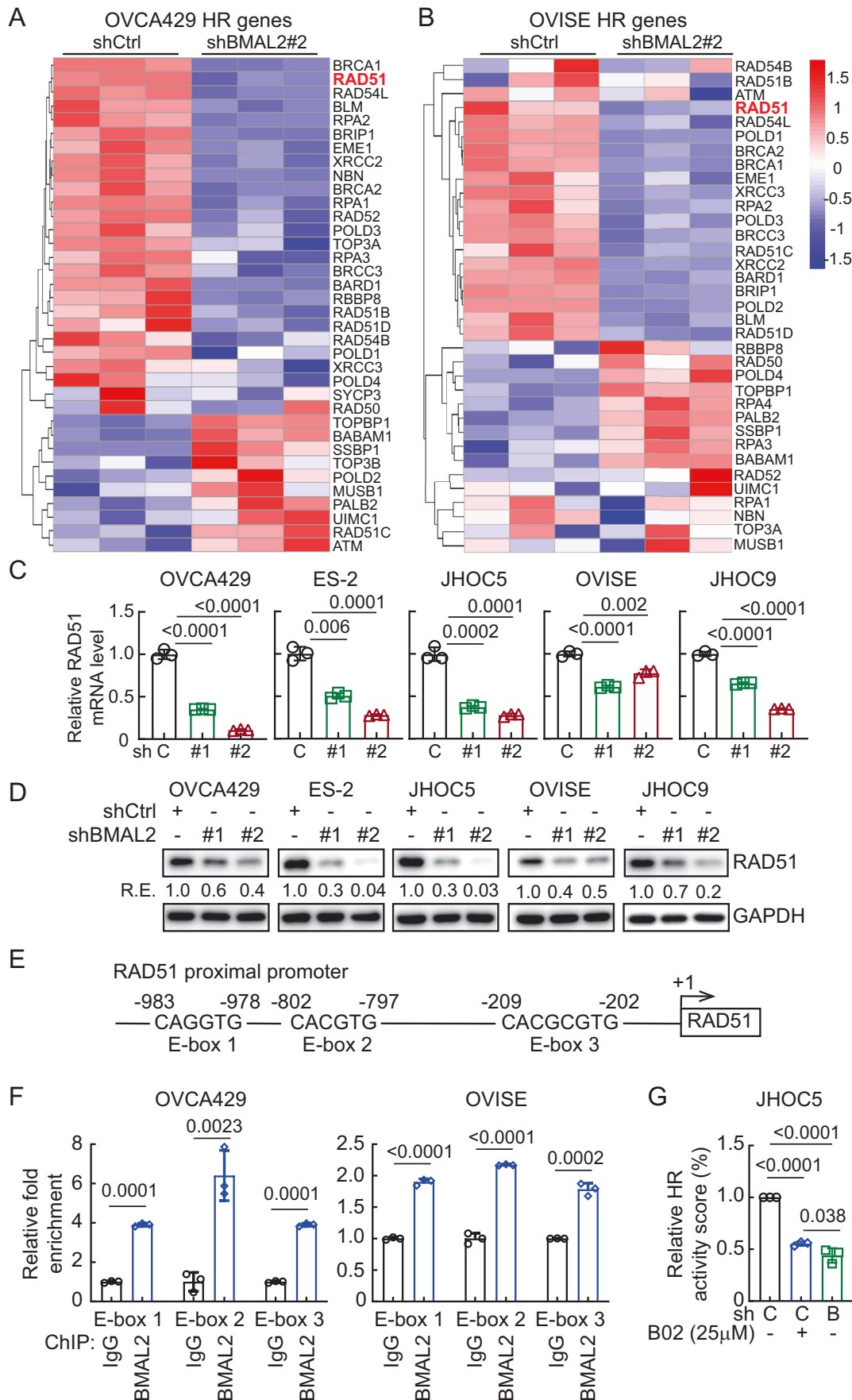

◀ **Figure EV4. BMAL2 depletion downregulates genes in the homologous recombination (HR) pathway, including RAD51.**

(A, B) Heatmap of HR genes of the KEGG pathway (hsa03440) in OVCA429 (A) and OVISE (B) cells without or with BMAL2 depletion. (C) qRT-PCR analysis of RAD51 level in OCCC cells without (shCtrl) or with BMAL2 depletion (shBMAL2#1 or #2). RNA18S5 was used as an internal control. Three independent experiments were performed, and data were means ± SD from one representative experiment ($n = 3$). ***$P < 0.001$; ****$P < 0.0001$. Significant differences are based on an unpaired $t$-test. (D) IB of RAD51 protein expression with GAPDH as a loading control in shCtrl or shBMAL2 OCCC cells. Blots shown are from one representative experiment of three replicates. RE relative expression. (E) Diagram shows three putative E-boxes on the *RAD51* promoter predicted using the EPD eukaryotic promoter database. (F) ChIP-qPCR analysis of BMAL2 on the *RAD51* promoter E-box regions. Three independent experiments were performed, and data are means ± SD from one representative experiment with significant differences detected by an unpaired $t$-test. ***$P < 0.001$; ****$P < 0.0001$. (G) HR activity of shCtrl or shBMAL2 JHOC5 cells, assessed 72 h after adenovirus infection. Data were shown as mean ± SD with $P$ value based on one-way ANOVA test ($n = 3$). Source data are available online for this figure.

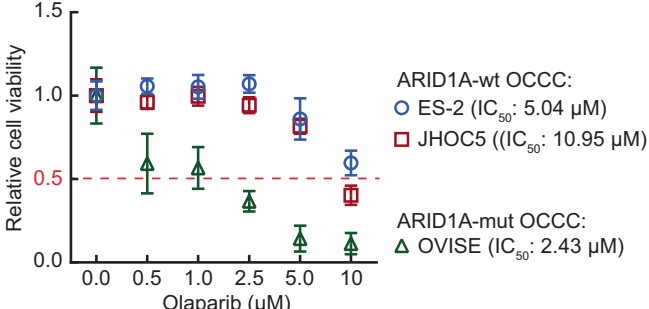

**Figure EV5.  ARID1A-mut OCCC cells are more sensitive to PARPi than ARID1A-wt OCCC cells.**

Cell viability assays using ES-2, JHOC5 and OVISE cells treated with vehicle (0), 0.5, 1, 2.5, 5, or 10 μM Olaparib. Data were shown as mean ± SD ($n = 3$). The half-maximal inhibitory concentration ($IC_{50}$) of Olaparib for each cell line is indicated. Source data are available online for this figure.

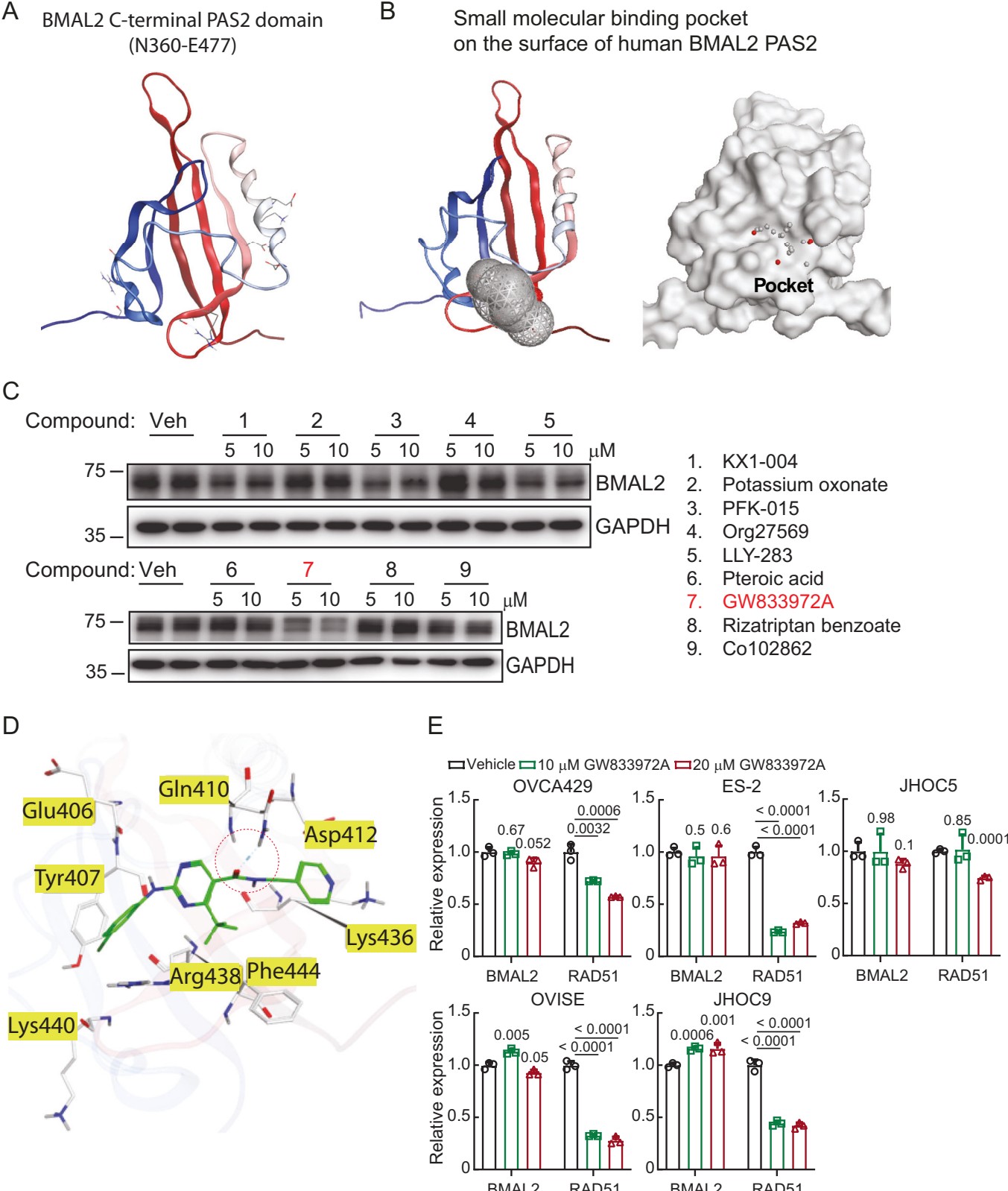

**Figure EV6.   Virtual screening of potential small compounds for BMAL2 inhibition.**

(A) 3D structure of human BMAL2 PAS2 domain (2KDK). The N-terminal to C-terminal structure was colored from blue to red. (B) The small molecular binding pocket on the surface of human BMAL2 PAS2 domain was identified using MOE-SiteFinder, where the pocket was served as the docking area for the virtual screening. The docking area was defined by a docking box with the length, width and height of 19.67 Å × 20.33 Å × 16.67 Å respectively, and the total volume was 6664 Å$^3$, where the inner contour volume was 2614 Å$^3$. (C) Left panel: IB of BMAL2 protein with GAPDH as a loading control in ES-2 cells treated with vehicle (DMSO), 5 or 10 μM selected compounds. Blots shown are from one representative experiment of three replicates. Right panel: The top nine bioactive compounds with low $K_d$ value used for evaluation. (D) Binding modes of GW833972A to human BMAL2. BMAL2-compound complex, in which the interaction of GW833972A with the contact residues of the BMAL2 PAS2 domain were shown. (E) qRT-PCR analysis of BMAL2 and RAD51 levels in OCCC cells treated with vehicle, 10 or 20 μM GW833972A. RNA18S5 was used as an internal control. Three independent experiments were performed, and data were means ± SD from one representative experiment ($n = 3$). Significant differences are based on an unpaired *t*-test. Source data are available online for this figure.

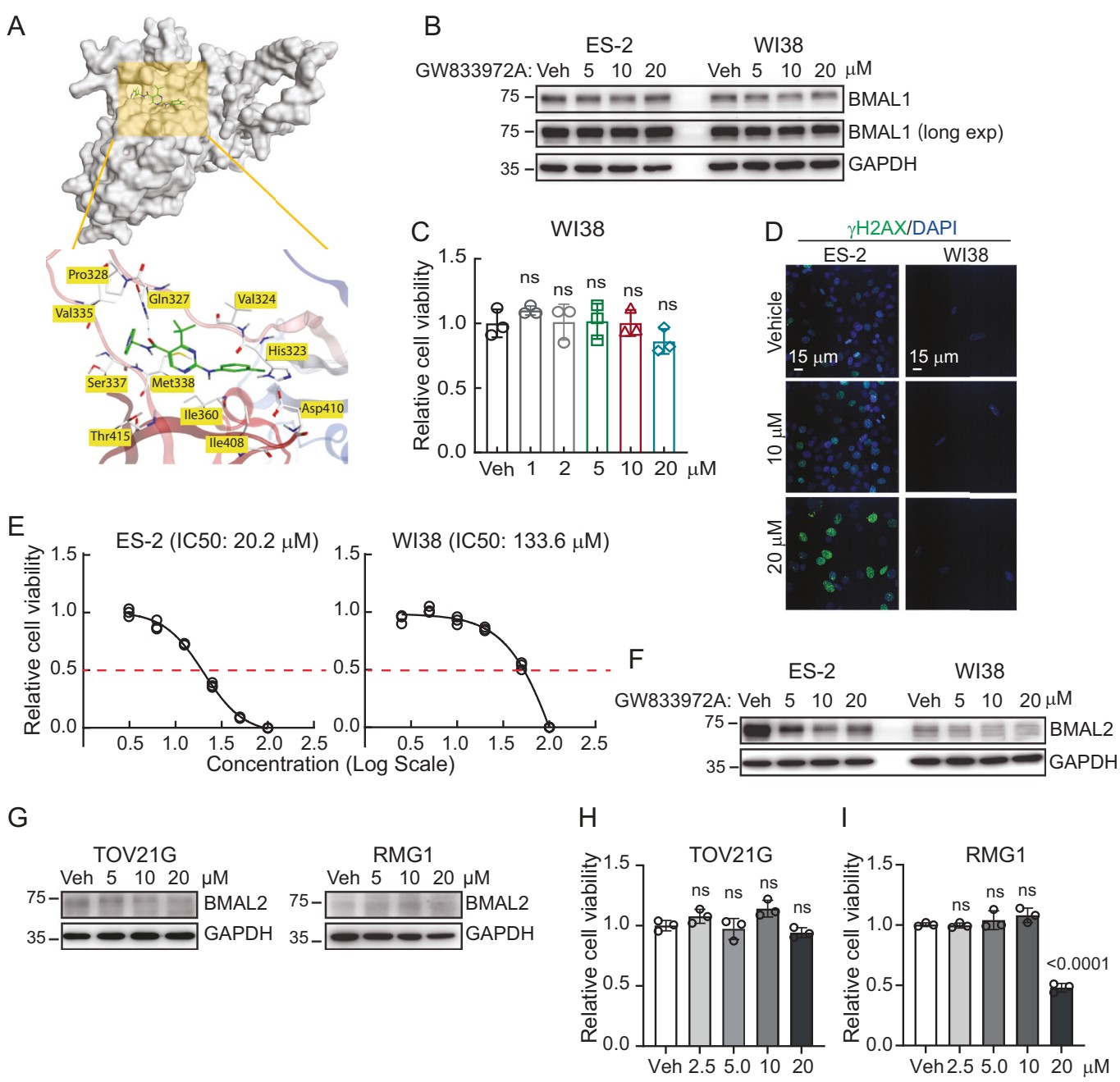

**Figure EV7. GW833972A can effectively target BMAL2- expressing cancer cells while having low adverse effects on normal cells.**

(A) Binding mode of GW833972A to human BMAL1. BMAL1-compound complex, in which the BMAL1 core circadian modulator (CCM) binding pocket in the PAS-B domain was shown as a molecular surface, and GW833972A was shown in green. Interaction of GW833972A to the contact residues of BMAL1 were shown. (B) IB of BMAL1 protein with GAPDH as a loading control in ES-2 and WI38 cells treated with vehicle (DMSO), 5, 10, or 20 μM GW833972A. Blots shown are from one representative experiment of three replicates. (C) Cell viability assays using WI38 cells treated with vehicle (DMSO), 5, 10, or 20 μM GW833972A. Data were shown as mean ± SD with P value based on unpaired t-test (n = 3). ns not significant. The experiments were repeated three times. (D) Representative γH2AX staining of ES-2 and WI38 cells treated with vehicle (DMSO), 10 or 20 μM GW833972A. Scale bar indicates 15 μm. (E) Dose-response curves for the assessment of cell viability in ES-2 and WI38 cells treated by GW833972A from 0 to 100 μM. The curves were plotted with log10 [GW833972A (μM)]. IC50 for each cell line was indicated. (F) IB of BMAL2 protein with GAPDH as a loading control in ES-2 and WI38 cells treated with vehicle (DMSO), 5, 10, or 20 μM GW833972A. Blots shown are from one representative experiment of three replicates. (G) IB of BMAL2 protein with GAPDH as a loading control in TOV21G and RMG1 cells treated with vehicle (DMSO), 5, 10, or 20 μM GW833972A. Blots shown are from one representative experiment of three replicates. (H, I) Cell viability assays using TOV21G (H) and RMG1 (I) cells treated with vehicle (DMSO), 5, 10, or 20 μM GW833972A. Data were shown as mean ± SD with P value based on unpaired t-test (n = 3). ns not significant. The experiments were repeated three times. Source data are available online for this figure.

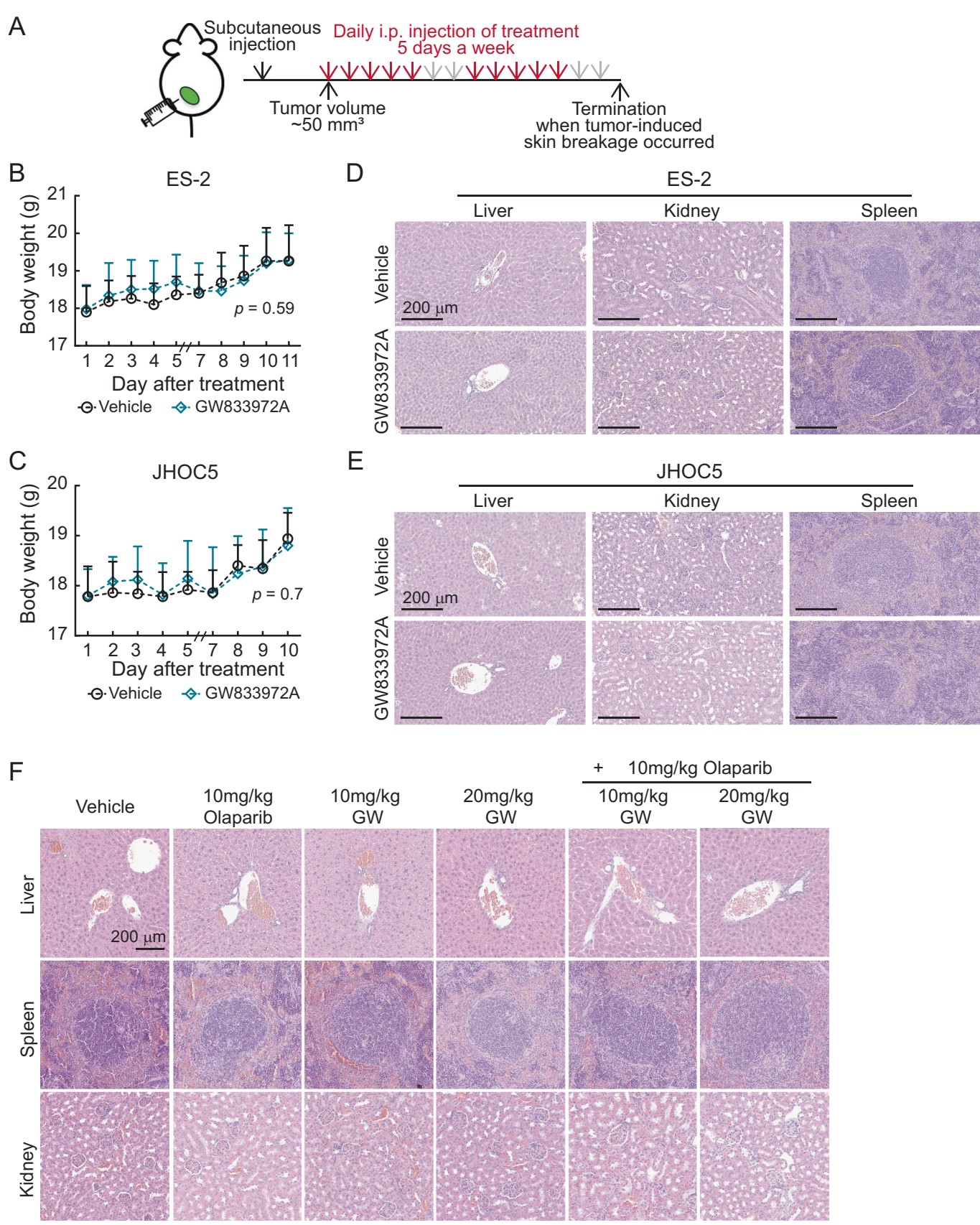

◀ **Figure EV8.   Subcutaneous xenograft models using ES-2 and JHOC5 cells.**

(A) Diagram of the procedure used for subcutaneous xenografts. $2 \times 10^6$ ES-2 or $5 \times 10^6$ JHOC5 cells mixed with Matrigel in a 1:1 ratio were subcutaneously injected into the lower flanks of the mouse. When tumors reached 50 mm$^3$, mice were intraperitoneally injected with either vehicle or 10 mg/kg GW833972A daily for 5 days a week. Mice were euthanized when tumors in the control group reached a 2-cm diameter or when the tumor-induced skin breakage occurred in the vehicle control group. (B) Subcutaneous xenograft model in NUDE mice using ES-2 cells with or without 10 mg/kg GW833972A treatment. Five mice were used for each group, and data were shown as mean ± SD ($n = 5$). Nonlinear regression (curve fit) analysis was used to test for significant differences between the body weight curve of the vehicle control and GW833972A-treated group ($P$ value is indicated). (C) Subcutaneous xenograft model in NUDE mice using JHOC5 cells with or without 10 mg/kg GW833972A treatment. Five mice were used for each group, and data were shown as mean ± SD ($n = 5$). Data formatting is as described for (B). (D) Representative images of hematoxylin and eosin (H&E) staining using liver, kidney, and spleen tissue sections from mice bearing ES-2-derived tumors. Scale bars indicate 200 µm. (E) Representative images of hematoxylin and eosin (H&E) staining using liver, kidney and spleen tissue sections from mice bearing JHOC5-derived tumors. Scale bars indicate 200 µm. (F) Representative images of hematoxylin and eosin (H&E) staining using liver, kidney, and spleen tissue sections from mice bearing ES-2 derived tumors treated with either vehicle, 10 mg/kg Olaparib, 10 mg/kg or 20 mg/kg GW833972A (GW), or GW833972A-Olaparib combination. Scale bars indicate 200 µm. Source data are available online for this figure.

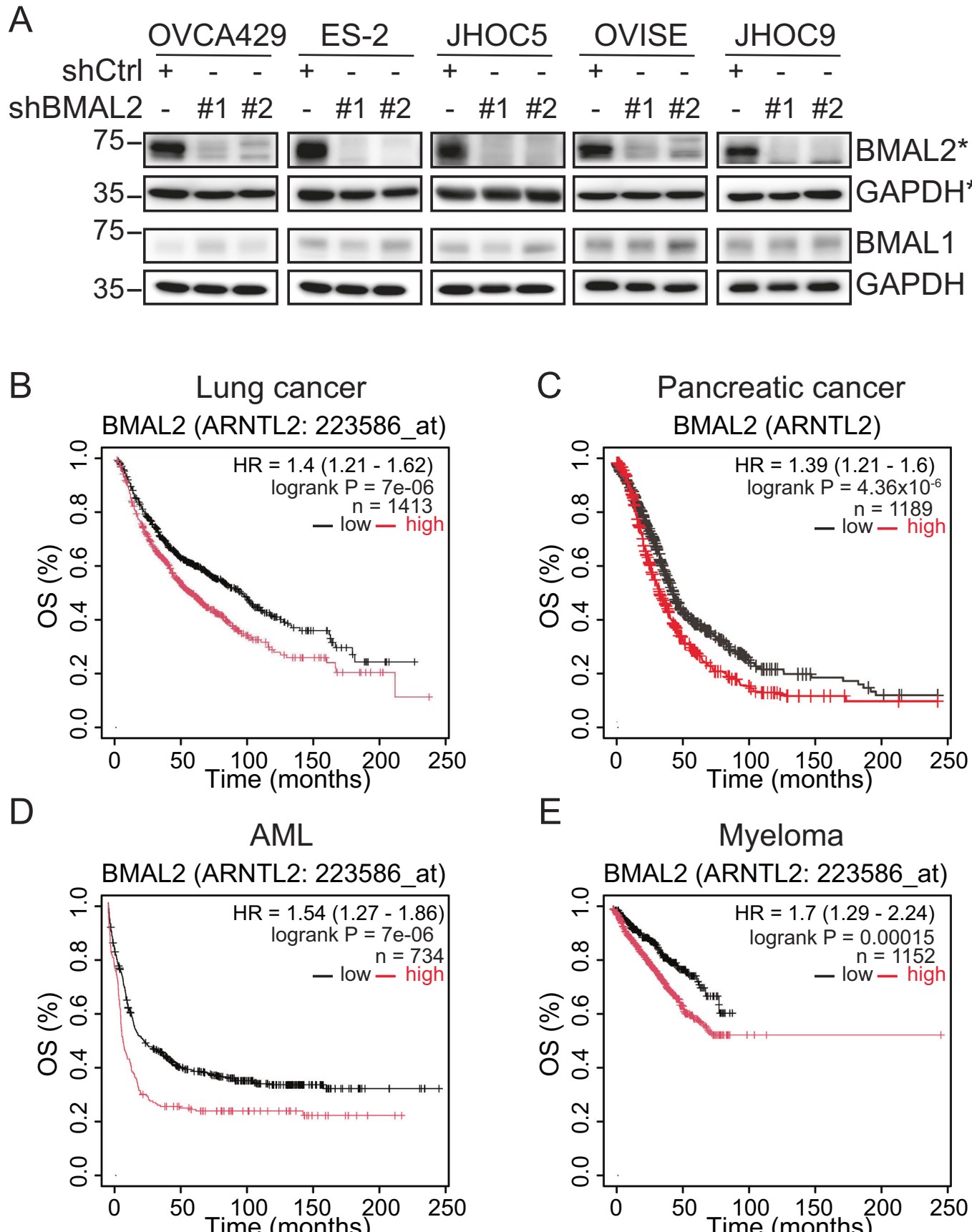

◀ **Figure EV9. High BMAL2 expression is correlated with poor clinical outcomes in several cancer types other than OC.**

(A) Depletion of BMAML2 did not affect BMAL1 level in OCCC cells. IB of BMAL1 and BMAL2 protein levels with GAPDH as a loading control in OCCC cell lines without (shCtrl) or with BMAL2 depletion (shBMAL2#1 or #2). Blots shown are from one representative experiment of three replicates. *, BMAL2 blots presented here were from Fig. 2B. (B–E) Kaplan–Meier overall survival (OS) analysis of lung cancer (B), pancreatic cancer (C), AML (D), and myeloma (E) cancer patients grouped by BMAL2 expression. The BMAL2 high group is indicated by a red line and the BMAL2 low group is indicated by a black line. The P value was determined by the log-rank test. The results shown here are based upon data generated by the KM plotter (https://kmplot.com/analysis/). Source data are available online for this figure.

