## [Peer Review File · EMBO Molecular Medicine]

BMAL2 is a druggable target for ovarian clear cell carcinoma (OCCC)

Grace Tan, Pei-Yi Lin, Li-Tzu Cheng, Yu-Sheng Tsai Yuan, Shih-Han Huang, Chen-Hsin Yu, Chao-Tsen Chen, Peter (Hung-Yuan) Chi, and Wendy W. Hwang-Verslues

Corresponding author(s): Wendy W. Hwang-Verslues (wendyhv@gate.sinica.edu.tw)

Review Timeline:

Submission Date:	30th Oct 25
Editorial Decision:	17th Nov 25
Revision Received:	7th Feb 26
Editorial Decision:	10th Mar 26
Revision Received:	13th Mar 26
Accepted:	19th Mar 26

Editor: Lise Roth

Transaction Report:

17th Nov 2025

Dear Dr. Hwang-Verslues,

Thank you for submitting your manuscript to EMBO Molecular Medicine. We have now received feedback from the three reviewers who agreed to evaluate it. As you will see from the reports below, the reviewers acknowledge the quality, novelty and potential translational interest of the findings but nevertheless raise a few major concerns. If you feel you can address the reviewers' comments satisfactorily, you may wish to submit a revised version of your manuscript. Please note that, following further consultation with the referees, we have agreed that, although further investigation of BMAL1 is expected, in vivo experiments on this matter will NOT be required.

Addressing the reviewers' concerns in full will be necessary for further considering the manuscript in our journal, and acceptance of the manuscript will entail a second round of review. EMBO Molecular Medicine encourages a single round of revision only and therefore, acceptance or rejection of the manuscript will depend on the completeness of your responses included in the next, final version of the manuscript. For this reason, and to save you from any frustrations in the end, I would strongly advise against returning an incomplete revision.

We are expecting your revised manuscript within three months, if you anticipate any delay, please contact us.

We require:

Additional information on source data and instruction on how to label the files are available

4) A .docx formatted letter INCLUDING the reviewers' reports and your detailed point-by-point responses to their comments. As part of the EMBO Press transparent editorial process, the point-by-point response is part of the Review Process File (RPF), which will be published alongside your paper.

5) A complete author checklist, which you can download from our author guidelines (<https://www.embopress.org/page/journal/17574684/authorguide#submissionofrevisions>). Please insert information in the checklist that is also reflected in the manuscript. The completed author checklist will also be part of the RPF.

6) All Materials and Methods need to be described in the main text using our 'Structured Methods' format. According to this format, the Methods section includes a Reagents and Tools Table (listing key reagents, experimental models, software and relevant equipment and including their sources and relevant identifiers) followed by a Methods and Protocols section describing the methods, ideally using a step-by-step protocol format. The aim is to facilitate adoption of the methodologies across labs. Please download and fill our Reagents and Tools Table template (.docx), which you can find in our author guidelines: <https://www.embopress.org/page/journal/14693178/authorguide#structuredmethods>.

7) Please note that all corresponding authors are required to supply an ORCID ID for their name upon submission of a revised manuscript.

8) It is mandatory to include a 'Data Availability' section after the Materials and Methods. Before submitting your revision, primary datasets produced in this study need to be deposited in an appropriate public database, and the accession numbers and database listed under 'Data Availability'. Please remember to provide a reviewer password if the datasets are not yet public (see <https://www.embopress.org/page/journal/17574684/authorguide#dataavailability>).

9) For data quantification: please specify the name of the statistical test used to generate error bars and P values, the number (n) of independent experiments (specify technical or biological replicates) underlying each data point and the test used to calculate p-values in each figure legend. The figure legends should contain a basic description of n, P and the test applied. Graphs must include a description of the bars and the error bars (s.d., s.e.m.). Please provide exact p values.

10) Our journal encourages inclusion of *data citations in the reference list* to directly cite datasets that were re-used and obtained from public databases. Data citations in the article text are distinct from normal bibliographical citations and should directly link to the database records from which the data can be accessed. In the main text, data citations are formatted as follows: "Data ref: Smith et al, 2001" or "Data ref: NCBI Sequence Read Archive PRJNA342805, 2017". In the Reference list, data citations must be labeled with "[DATASET]". A data reference must provide the database name, accession number/identifiers and a resolvable link to the landing page from which the data can be accessed at the end of the reference. Further instructions are available at .

11) We replaced Supplementary Information with Expanded View (EV) Figures and Tables that are collapsible/expandable online. EV Figures should be cited as 'Figure EV1, Figure EV2' etc... in the text and their respective legends should be included in the main text after the legends of regular figures.

12) The paper explained: EMBO Molecular Medicine articles are accompanied by a summary of the articles to emphasize the major findings in the paper and their medical implications for the non-specialist reader. Please provide a draft summary of your article highlighting

13) Author contributions: CRediT has replaced the traditional author contributions section because it offers a systematic machine readable author contributions format that allows for more effective research assessment. Please remove the Authors Contributions from the manuscript and use the free text boxes beneath each contributing author's name in our system to add specific details on the author's contribution. More information is available in our guide to authors.

Please also suggest a visual abstract to illustrate your article as a PNG file 550 px wide x 300-600 px high. A cropped portion of this image will serve as thumbnail for the table of content on our webpage.

16) As part of the EMBO Publications transparent editorial process initiative (see our Editorial at <http://embomolmed.embopress.org/content/2/9/329>), EMBO Molecular Medicine will publish online a Review Process File (RPF) to accompany accepted manuscripts.

In the event of acceptance, this file will be published in conjunction with your paper and will include the anonymous referee reports, your point-by-point response and all pertinent correspondence relating to the manuscript. Let us know whether you agree with the publication of the RPF and as here, if you want to remove or not any figures from it prior to publication. Please note that the Authors checklist will be published at the end of the RPF.

I look forward to receiving your revised manuscript.

Yours sincerely,

Lise Roth

***** Reviewer's comments *****

Referee #1 (Comments on Novelty/Model System for Author):

The study is technically sound and generally well executed. The authors use multiple complementary approaches-cell lines, xenograft models, and biochemical assays (CETSA, DARTS, CHX chase)-to support that BMAL2 regulates RAD51 transcription, thereby maintaining homologous recombination (HR) and genomic integrity in ovarian clear cell carcinoma (OCCC). Identification of GW833972A as a small molecule that induces BMAL2 degradation adds translational novelty.

However, one conceptual weakness concerns why ARID1A-wildtype (wt) OCCC is highlighted as the key disease context. The current data mostly compare ARID1A-wt and -mut cell lines, but the rationale for prioritizing the wt subgroup over others is not sufficiently substantiated by clinical evidence. The manuscript would benefit from re-analysis of existing clinical or transcriptomic datasets (e.g., TCGA, CSIOVDB, GSE...) to demonstrate that:

ARID1A-wt OCCC indeed shows poor prognosis and limited response to standard chemotherapy or PARP inhibitors, and BMAL2 expression is particularly enriched or prognostically relevant within the ARID1A-wt subset.

This would firmly justify the therapeutic emphasis on BMAL2 in ARID1A-wt tumors and clarify the model's clinical adequacy.

Referee #1 (Remarks for Author):

1. Figure 1 - Rationale and data representation

The initial motivation for focusing on BMAL2 in OCCC is not entirely convincing.

Most data in Fig. 1 (TCGA, KM-plot) represent high-grade serous carcinoma (HGSC) rather than OCCC.

The authors should strengthen the rationale by including clinical or transcriptomic analyses specific to OCCC, ideally stratified by ARID1A status.

In Fig. 1F, BMAL2 IHC compares OCCC with normal ovary.

Since OCCC typically arises from endometriosis or endometrioid epithelium, a comparison with ectopic or eutopic endometrium would be pathobiologically more appropriate.

Suggest adding a figure or supplementary panel showing clinical outcomes of ARID1A-wt vs. ARID1A-mut OCCC to reinforce the unmet clinical need.

2. Figure 2 - In vivo data presentation

The shBMAL2 xenograft data (Fig. 2G-I) convincingly show growth suppression, but only representative images are presented.

Please provide all individual tumor photographs or animal-by-animal growth curves to eliminate selection bias.

The Ki67 IHC in Fig. 2J-L and EV1D supports reduced proliferation; consider including quantitative scoring (H-score or percentage).

3. Figure 3 - PARPi combination and synergy analysis

BMAL2 depletion reduces RAD51 and HR capacity, logically implying increased sensitivity to PARP inhibition.

However, the current data (Figs 3D-E) show only additive or ceiling-type inhibition.

The authors should quantitatively evaluate synergy using Bliss independence, Loewe additivity, or ZIP analysis, ideally across a 2D dose matrix.

Clarify whether BMAL2 knockdown alone already induces near-maximal cytotoxicity, masking potential synergy.

If combination benefits are minimal, the therapeutic positioning (monotherapy vs. combination) should be explicitly discussed.

4. Figure 4 - Biochemical validation of compound binding

CETSA and DARTS assays appropriately demonstrate target engagement of GW833972A with BMAL2; these are standard biochemical validations and not strictly mandatory but do strengthen the claim.

The CHX chase experiment (Fig. 4E-F) is intended to assess BMAL2 protein half-life, showing that GW833972A accelerates degradation.

The authors should clarify this rationale explicitly in the text.

Mechanistic depth could be improved by identifying the degradation pathway (e.g., proteasome dependence, ubiquitination), which remains speculative.

5. Figure 5-6 - Therapeutic interpretation

In Figs 5E-F, low-dose GW833972A enhances Olaparib sensitivity, but at high dose (20 μ M), GW833972A alone achieves maximal inhibition, and no further effect of Olaparib is observed.

This suggests a strong single-agent effect rather than synergy. Please clarify this interpretation.

The in-vivo study (Fig. 6) demonstrates efficacy of GW833972A monotherapy only.

Including a combination arm (GW + Olaparib) would better support the translational claim.

6. Discussion

The first two paragraphs mainly repeat background material suited for the Introduction.

Consider shortening or moving these and focusing the Discussion on how BMAL2-RAD51 regulation links to existing DNA-repair literature and how BMAL2 degradation differs mechanistically from other synthetic-lethal strategies (e.g., ARID1A-ATR, ARID1A-EZH2).

Strengthen the integration of your own findings with prior reports on BMAL1/BMAL2 and circadian-independent DNA-repair regulation.

7. Nomenclature and consistency errors

Compound name inconsistency:

The compound is referred to as GW833972A throughout the main text but as GW833982A in the Supplementary "Detailed structural characterization" section.

Please correct this discrepancy and ensure that the NMR/HRMS data correspond to the actual compound used in biological assays.

Table EV1 caption error:

The caption incorrectly mentions "shMEX3A"; this should be "shBMAL2."

Verify that all figure and table numbering corresponds correctly between main and expanded view files.

Referee #2 (Comments on Novelty/Model System for Author):

This paper proposes BMAL2 as a therapeutic target for ARID1A-wildtype ovarian clear cell carcinoma (OCCC), a subgroup lacking effective treatments. The authors show that BMAL2 supports DNA repair by regulating RAD51 expression, and its depletion triggers double-strand breaks and cell death. They identify GW833972A, a cannabinoid receptor agonist, as a compound that binds BMAL2, promotes its degradation, and suppresses tumour growth in xenograft models. This work highlights BMAL2's role in maintaining genomic integrity and suggests that targeting BMAL2 could offer a new strategy for treating ARID1A wild type OCCC. This manuscript comprises an interesting and thorough study. Generally, this study has been performed to a high standard with the appropriate cell lines and controls. The data in the manuscript are interesting and novel, generally the experimental methods are sound, and the figures are of high quality.

Referee #2 (Remarks for Author):

This paper proposes BMAL2 as a therapeutic target for ARID1A-wildtype ovarian clear cell carcinoma (OCCC), a subgroup lacking effective treatments. The authors show that BMAL2 supports DNA repair by regulating RAD51 expression, and its depletion triggers double-strand breaks and cell death. They identify GW833972A, a cannabinoid receptor agonist, as a compound that binds BMAL2, promotes its degradation, and suppresses tumour growth in xenograft models. This work highlights BMAL2's role in maintaining genomic integrity and suggests that targeting BMAL2 could offer a new strategy for treating ARID1A wild type OCCC. This manuscript comprises an interesting and thorough study. Generally, this study has been performed to a high standard with the appropriate cell lines and controls. The data in the manuscript are interesting and novel, generally the experimental methods are sound, and the figures are of high quality.

Considering the extensive data in this study, I support the publication of this manuscript in EMBO Molecular Medicine subject to

the below points being sufficiently addressed:

1. The text on Figure 1A needs to be enlarged as it's currently illegible.
2. Presumably, the authors chose the cell lines expressing the highest levels of BMAL2 in Figure 2a to perform the rest of the experiments on? If so, please state this in the text on page 5. It would have been interesting to see whether the BMAL2 ShRNA could still further inhibit cell growth even in the low expressing cell lines or whether it has no additional effect. Perhaps BMAL2 levels could serve as a biomarker for GW833972A.
3. Many of the blots in the supplementary figures are very blurry and low resolution (this may be an artifact of the uploading process) but these need to be better quality so they can be adequately assessed.
4. Several of the figure legends state 'The experiments were repeated 3 times.' Please can the authors clarify whether the data presented is representative of the data of the 3 repeats or only one experiment.
5. BMAL2 is not defined as an essential gene as it's functions are mostly compensated by BMAL1. The authors do touch on BMAL1 levels being low in ovarian cancer and that seems to be the case in their cell lines, but I do wonder if what they are seeing is synthetic lethality in cell lines that are low in BMAL1 already, since BMAL1 and BMAL2 can compensate for each other. The authors should over-express BMAL1 in their cell lines (or select a cell line with high BMAL1 and BMAL2 levels) to assess whether BMAL1 can protect against BMAL2 KD and GW833972A. This should also be addressed in the discussion.
6. The authors state that 'BMAL2 knockdown did not lead to increase in γ H2AX foci in HEYA8 and KURAMOCHI cells (Fig. EV2D), suggesting that BMAL2-mediated DNA integrity maintenance is an OCCC subtype-specific regulatory mechanism'. This is a curious observation, but the authors need to show the BMAL1 and BMAL2 levels and extent of the BMAL2 knockdown in the HEYA8 and KURAMOCHI cells to enable a comparison with the OCCC cells. It may just be in tumour cells with high BMAL2 levels (and potentially low BMAL1 levels) rather than specific to OCCC. Therefore, this statement should probably be toned down.
7. Figure 3D-E, please add the olaparib dose to the figure legend. There is also no figure legend for Figure 3E.
8. A chemist/molecular modelling expert should review the screening section as this is not my primary expertise, but it seems unlikely that 8/9 compounds from the molecular docking screen would affect BMAL2 stability. It is generally estimated that 10-30% of hits from screens will bind and even less will have a functional effect on the protein. The authors need to provide the full high-resolution blots as the current blots show very blurry bands and they need to show that the GAPDH and BMAL2 bands are from the same membrane. EV5 should also be a figure in the main manuscript as it is an important part of the manuscript.
9. The authors also need to determine whether GW833972A would bind to BMAL1 as the PAS2 domains in BMAL1 and BMAL2 are highly homologous. This could be addressed via modelling (if the BMAL1 structure is available) or the authors could look at BMAL1 protein levels to determine if it is also down regulated.
10. The effect of BMAL2 knockdown or GW833972A on normal human cells has not been explored at all in this study, so targeting BMAL2 could be toxic to non-cancerous human cells. The authors should test BMAL2 knockdown and GW833972A in human primary fibroblasts to assess the effects on cell growth and viability.
11. Discussion: Since Rad51 inhibitors are already in clinical trials the authors need to discuss how BMAL2 inhibitors may be a different approach, i.e. will they show any advantage over Rad51 inhibitors?

Minor typos:

Page 7: Olaparib should be olaparib.

Referee #3 (Comments on Novelty/Model System for Author):

Experiments were performed in triplicates and appropriate statistical analysis was performed.

Identification of BMAL2 as a druggable oncogene in ARID1A wild-type ovarian clear cell carcinoma is new and medically relevant.

The experiments were performed in vitro (using various OCCC lines) and in mice using xenografts.

Referee #3 (Remarks for Author):

This manuscript focuses on ovarian clear cell carcinoma (OCCC), which has a very poor survival rate and shows chemoresistance, especially in the case of ARID1A wild-type background (50% of OCCC cases). The authors identified BMAL2

as an oncogene in OCCC, which promotes homologous recombination (HR) repair by sustaining the expression of HR genes such as RAD51. BMAL2 is upregulated in ovarian cancer, especially in OCCC. Using structure-based screening, they identified a compound that can block BMAL2. They show that depletion or inhibition of BMAL2 reduces cancer cell proliferation in vitro and in a xenograft model, and that it can synergize with PARP inhibition. The compound does not have the same effects on normal fibroblasts, suggesting that it wouldn't have severe side effects. Overall, the study is medically relevant and well-conducted, the manuscript is clearly written. There are a few missing experiments or explanations that are required to consolidate the overall quality, as indicated below.

1) Figure 1F: Please explain at least in the figure legend why PAX8 was used in IHC.

2) Figure 2A: There is a high variability in BMAL2 expression across OCCC cell lines (some with very high but some with very low BMAL2). Please include a normal ovary cell line and a non-OCCC ovarian cancer cell line for better comparison.

3) Figure 2B-F: RMG1 and TOV21G cell lines should be included as well because they express lower levels of BMAL2, so presumably the effect of BMAL2 depletion on cell proliferation would be weaker. One of these two cell lines should also be tested in mouse experiments (Figure 2G-L).

4) Can you please indicate that experiments in Figure 3 and EV2 were performed in untreated cells (without DNA-damaging agents).

5) The effect of GW833972A (Figure 5) should be tested in low BMAL2-expressing cell lines like RMG1 and TOV21G to presumably confirm that they would not respond to this compound.

Point-to-point response

We thank the reviewers for their positive feedback and constructive comments that allowed us to greatly improve our study. Please find the point-to-point response below.

Referee #1 (Comments on Novelty/Model System for Author):

The study is technically sound and generally well executed. The authors use multiple complementary approaches-cell lines, xenograft models, and biochemical assays (CETSA, DARTS, CHX chase)-to support that BMAL2 regulates RAD51 transcription, thereby maintaining homologous recombination (HR) and genomic integrity in ovarian clear cell carcinoma (OCCC). Identification of GW833972A as a small molecule that induces BMAL2 degradation adds translational novelty.

However, one conceptual weakness concerns why ARID1A-wildtype (wt) OCCC is highlighted as the key disease context. The current data mostly compare ARID1A-wt and -mut cell lines, but the rationale for prioritizing the wt subgroup over others is not sufficiently substantiated by clinical evidence. The manuscript would benefit from re-analysis of existing clinical or transcriptomic datasets (e.g., TCGA, CSIOVDB, GSE...) to demonstrate that:

ARID1A-wt OCCC indeed shows poor prognosis and limited response to standard chemotherapy or PARP inhibitors, and

BMAL2 expression is particularly enriched or prognostically relevant within the ARID1A-wt subset.

This would firmly justify the therapeutic emphasis on BMAL2 in ARID1A-wt tumors and clarify the model's clinical adequacy.

Response: We thank the reviewer for the insights. We agree with the reviewer that adding data from clinical or transcriptomic datasets specific to OCCC, particularly containing ARID1A status, could strengthen our rationale significantly. Please see the point-to-point response to the comment #1 below.

We apologize if our writing was unclear. We did not intend to claim that BMAL2 expression is particularly enriched in ARID1A-wt or that targeting BMAL2 is a good strategy to treat only ARID1A-wt OCCC. As described in our manuscript, BMAL2 expression can be observed in both ARID1A-wt and ARID1A-mut OCCC, and BMAL2 depletion in both ARID1A-wt and ARID1A-mut cancer cells can effectively inhibit tumorigenesis. We highlighted ARID1A-wt OCCC because ARID1A-mut OCCC is the focus of the current synthetic lethal strategies; however, there is no targeted therapy available for ARID1A-wt OCCC (~50% of the OCCC cases). In the revised manuscript, we add new data and clarify that targeting BMAL2 by GW833972A can effectively inhibit tumorigenic ability of both ARID1A-wt and ARID1A-mut OCCC. We then explain why our study can provide a therapeutic strategy, particularly for ARID1A-wt OCCC, in the discussion section. In the revised manuscript, we modified the title from "BMAL2 is a druggable target for ARID1A-wildtype ovarian clear cell carcinoma (OCCC)" to "BMAL2 is a druggable target for ovarian clear cell carcinoma (OCCC)".

Referee #1 (Remarks for Author):

1. Figure 1 - Rationale and data representation

The initial motivation for focusing on BMAL2 in OCCC is not entirely convincing. Most data in Fig. 1 (TCGA, KM-plot) represent high-grade serous carcinoma (HGSC) rather than OCCC. The authors should strengthen the rationale by including clinical or transcriptomic analyses specific to OCCC, ideally stratified by ARID1A status. In Fig. 1F, BMAL2 IHC compares OCCC with normal ovary. Since OCCC typically arises from endometriosis or endometrioid epithelium, a comparison with ectopic or eutopic endometrium would be pathobiologically more appropriate. Suggest adding a figure or supplementary panel showing clinical outcomes of ARID1A-wt vs. ARID1A-mut OCCC to reinforce the unmet clinical need.

Response: We thank the reviewer for the comment. We agree with the reviewer that adding data from clinical or transcriptomic datasets specific to OCCC, particularly containing ARID1A status, could strengthen our rationale significantly. Unfortunately, most studies focus on high-grade serous ovarian carcinoma. Only a few emphasize OCCC. These OCCC studies often have small sample sizes or incomplete information for their mutation status, transcriptomic information, treatment response or clinical outcome. Thus far, we only found one large scale OCCC study that produced data suitable for such analysis (421 OCCC tumor samples with targeted deep sequencing data specific to putative OCCC driver genes, including ARID1A ((Bolton *et al*, 2022) Clin Cancer Res 28 (22): 4947). Of the 421 tumor samples in that study, 211 samples have whole transcriptome sequencing data and part of them have information of clinical outcome. The authors reported that ARID1A-wt/TP53-mut OCCC tumors were associated with advanced stage disease and these patients had worse overall survival compared to those with ARID1A-mut OCCC tumors (HR = 1.72; 95% CI, 1.06–2.81; $P = 0.03$). This information is now included in the revised introduction on page 4.

To determine whether BMAL2 expression is prognostically related to ARID1A status of OCCC, we reanalyzed their raw data and extracted 164 cases that have information of ARID1A status and clinical outcome. ARID1A-wt patients ($n = 65$), regardless of their p53 status, tended to have worse overall survival than ARID1A-mut patients ($n = 99$). However, due to the small sample size, there is not a statistically significant difference ($P = 0.26$) (Rebuttal Fig. 1A). We further ranked these tumors by BMAL2 level, and compared the first (BMAL2-high) and fourth (BMAL2-low) quartiles both within and between ARID1A-wt ($n = 17$ for each group) and ARID1A-mut ($n = 25$ for each group) cases. No statistical significance was observed ($P = 0.56$) (Rebuttal Fig. 1B). While the sample size of this dataset was small, the results indicated that BMAL2 function may be independent from ARID1A status. After discussing with the editor, these results are included in the revised Figure EV2 and described on page 6 of the main text. However, if the reviewer thinks that these results are too preliminary to report, we will remove them from the revised supplementary information.

The reviewer's comment also prompted us to revise Figure 1 to demonstrate that BMAL2 is upregulated in OC. We added a new result using the CSIOVDB microarray gene expression database to compare BMAL2 expression in normal ovarian surface epithelium (OSE), stroma, fallopian tube epithelium (FTE) and ovarian tumor. A significant higher BMAL2 expression was found in the tumor samples compared to normal tissues (Rebuttal Fig. 1C). This new data is presented in the revised Fig. 1B.

We also added BMAL2 IHC data of 6 normal endometrioid epithelium samples as a normal control to compare BMAL2 level with OCCC cells (Rebuttal Fig. 1D). This new data is presented in the revised Fig. 1G.

Rebuttal Figure 1.

A and B. Kaplan–Meier plots for the survival probability following OCCC diagnosis stratified by mutational clusters defined by ARID1A mutation status (**A**) and BMAL2 expression level (**B**). The first (BMAL2-high) and fourth (BMAL2-low) quartiles both within and between ARID1A-wt and ARID1A-mut groups were compared. **C.** BMAL2 gene expression in normal ovarian surface epithelium (OSE), stroma, fallopian tube epithelium (FTE) and ovarian tumor analyzed using CSIOVDB database. The P value was determined by Mann Whitney test. **D.** (Left panel) Representative images of BMAL2 IHC staining using normal endometrioid epithelium (NEE). Black box indicates enlarged region. (Scale bars, 300 μm or 60 μm as shown in the images.) (Middle panel) Representative images of PAX8, a reliable clinical marker for OCCC (Chai et al, 2017; Gorbokon et al, 2024), and BMAL2 IHC staining using serial tumor sections from OCCC patients. Red boxes indicate enlarged tumor regions. Black boxes indicate enlarged non-tumor (NT) regions. (Scale bars, 200 μm or 60 μm as shown in the images.) (Right panel) Quantification and comparison of BMAL2 level in 6 NEE samples, and NT and OCCC tumor regions of 14 OCCC patient samples. Significant difference is based on paired t test.

2. Figure 2 - In vivo data presentation

The shBMAL2 xenograft data (Fig. 2G-I) convincingly show growth suppression, but only representative images are presented.

Please provide all individual tumor photographs or animal-by-animal growth curves to eliminate selection bias.

Response: All individual 3D ultrasound tumor images are now presented in the revised Figure EV1D (Rebuttal Fig. 2).

2-cont. The Ki67 IHC in Fig. 2J-L and EV1D supports reduced proliferation; consider including quantitative scoring (H-score or percentage).

Response: The reviewer may have missed this information. The quantification of Ki67 IHC is presented in Fig. 2J-L and the representative IHC images are presented in Fig. EV1E.

3. Figure 3 - PARPi combination and synergy analysis

BMAL2 depletion reduces RAD51 and HR capacity, logically implying increased sensitivity to PARP inhibition.

However, the current data (Figs 3D-E) show only additive or ceiling-type inhibition.

The authors should quantitatively evaluate synergy using Bliss independence, Loewe additivity, or ZIP analysis, ideally across a 2D dose matrix.

Clarify whether BMAL2 knockdown alone already induces near-maximal cytotoxicity, masking potential synergy.

If combination benefits are minimal, the therapeutic positioning (monotherapy vs. combination) should be explicitly discussed.

Response: As shRNA is hard to titrate, we performed the combination analysis using different doses of GW833972A and Olaparib (Rebuttal Fig. 3, revised Fig. 5). The combinatory effect of GW833972A and Olaparib was quantitatively evaluated using the Bliss independence model calculated by SynergyFinder 3.0. In ES-2 cells, the overall synergy score was 13.48 and the score of the most synergistic area (1.25-5 μ M Olaparib and 5-20 μ M GW833972A) was 24.45, indicating a strong synergistic effect when low doses of GW833972A and Olaparib were combined (revised Fig. 5G). In JHOC5 cells, the overall synergy score was 7.53 and the score of the most synergistic area (5-10 μ M Olaparib and 10-40 μ M GW833972A) was 10.26, indicating a moderate synergistic effect when both drugs were used (revised Fig. 5H). Together, these results indicate that GW833972A is effective by itself at high dose and can also be used at lower dosages to enhance the effectiveness of PARPi treatments in BMAL2-expressing OCC. The therapeutic positioning is discussed in the revised discussion section (page 13).

4. Figure 4 - Biochemical validation of compound binding

CETSA and DARTS assays appropriately demonstrate target engagement of GW833972A with BMAL2; these are standard biochemical validations and not strictly mandatory but do strengthen the claim.

The CHX chase experiment (Fig. 4E-F) is intended to assess BMAL2 protein half-life, showing that GW833972A accelerates degradation.

The authors should clarify this rationale explicitly in the text.

Mechanistic depth could be improved by identifying the degradation pathway (e.g., proteasome dependence, ubiquitination), which remains speculative.

Response: The rationale of these experiments is explained more clearly in the revised manuscript on page 9. We also conducted experiments to determine whether GW833972A-mediated BMAL2 degradation is proteasome dependent. MG132 treatment showed that inhibiting proteasome activity prevented GW833972A-induced degradation of BMAL2. These results suggested that GW833972A facilitates BMAL2 protein degradation via the proteasome system. These new results are presented in the revised Fig. 4G and 4H (Rebuttal Fig. 4).

5. Figure 5-6 - Therapeutic interpretation

In Figs 5E-F, low-dose GW833972A enhances Olaparib sensitivity, but at high dose (20 μM), GW833972A alone achieves maximal inhibition, and no further effect of Olaparib is observed.

This suggests a strong single-agent effect rather than synergy. Please clarify this interpretation.

The in-vivo study (Fig. 6) demonstrates efficacy of GW833972A monotherapy only.

Including a combination arm (GW + Olaparib) would better support the translational claim.

Response: (See also the response to point 3 above). We conducted cell-based combination analysis using different doses of GW833972A and Olaparib (Rebuttal Fig. 3). The data showed that when a low dose of GW833972A was used with PARPi, synergistic inhibition of cell viability was observed. When higher dose of GW833972A was used, monotherapy achieved substantial growth inhibition. When GW833972A at this high dose was used, combination with PARPi did not further increase its effectiveness. To validate these cell-based observations, a subcutaneous xenograft model using ES-2 was performed (Rebuttal Fig. 5). When tumors reached 50 mm³, mice were intraperitoneally injected daily with either vehicle, 10 mg/kg Olaparib, 10 mg/kg or 20 mg/kg GW833972A, or GW833972A-Olaparib combination until the experiment was terminated. Similar to the cell-based experiment, when GW833972A or Olaparib was applied individually, tumor growth was inhibited. When 10 mg/kg GW833972A was used in combination with Olaparib, tumor inhibition effect was further enhanced, but less than the sum of individual effect. Monotherapy using 20 mg/kg GW833972A achieved a more significant effect than 10 mg/kg GW833972A and Olaparib combined therapy. When the dose of GW833972A was increased, the combination effect with Olaparib was still detected but was smaller. Importantly, GW833972A treatment as monotherapy or in combination with Olaparib did not affect mouse body weight and did not cause abnormalities in the liver, kidney and spleen in these mice, suggesting low adverse side-effects of GW833972A. These results are presented in the revised Figure 6 (Rebuttal Fig. 5).

6. Discussion

The first two paragraphs mainly repeat background material suited for the Introduction. Consider shortening or moving these and focusing the Discussion on how BMAL2-RAD51 regulation links to existing DNA-repair literature and how BMAL2 degradation differs mechanistically from other synthetic-lethal strategies (e.g., ARID1A-ATR, ARID1A-EZH2).

Strengthen the integration of your own findings with prior reports on BMAL1/BMAL2 and circadian-independent DNA-repair regulation.

Response: We thank the reviewer for the suggestion. The discussion section was modified accordingly.

7. Nomenclature and consistency errors

Compound name inconsistency:

The compound is referred to as GW833972A throughout the main text but as GW833982A in the Supplementary "Detailed structural characterization" section.

Please correct this discrepancy and ensure that the NMR/HRMS data correspond to the actual compound used in biological assays.

Response: We apologize for this mistake. This error is corrected in the revised manuscript.

Table EV1 caption error:

The caption incorrectly mentions "shMEX3A"; this should be "shBMAL2."

Verify that all figure and table numbering corresponds correctly between main and expanded view files.

Response: This error is corrected in the revised manuscript.

Referee #2 (Comments on Novelty/Model System for Author):

This paper proposes BMAL2 as a therapeutic target for ARID1A-wildtype ovarian clear cell carcinoma (OCCC), a subgroup lacking effective treatments. The authors show that BMAL2 supports DNA repair by regulating RAD51 expression, and its depletion triggers double-strand breaks and cell death. They identify GW833972A, a cannabinoid receptor agonist, as a compound that binds BMAL2, promotes its degradation, and suppresses tumour growth in xenograft models. This work highlights BMAL2's role in maintaining genomic integrity and suggests that targeting BMAL2 could offer a new strategy for treating ARID1A wild type OCCC. This manuscript comprises an interesting and thorough study. Generally, this study has been performed to a high standard with the appropriate cell lines and controls. The data in the manuscript are interesting and novel, generally the experimental methods are sound, and the figures are of high quality.

Referee #2 (Remarks for Author):

This paper proposes BMAL2 as a therapeutic target for ARID1A-wildtype ovarian clear cell carcinoma (OCCC), a subgroup lacking effective treatments. The authors show that BMAL2 supports DNA repair by regulating RAD51 expression, and its depletion triggers double-strand breaks and cell death. They identify GW833972A, a cannabinoid receptor agonist, as a compound that binds BMAL2, promotes its degradation, and suppresses tumour growth in xenograft models. This work highlights BMAL2's role in maintaining genomic integrity and suggests that targeting BMAL2 could offer a new strategy for treating ARID1A wild type OCCC. This manuscript comprises an interesting and thorough study. Generally, this study has been performed to a high standard with the appropriate cell lines and controls. The data in the manuscript are interesting and novel, generally the experimental methods are sound, and the figures are of high quality.

Considering the extensive data in this study, I support the publication of this manuscript in EMBO Molecular Medicine subject to the below points being sufficiently addressed:

Response: We thank the reviewer for the positive feedback. Please find the point-to-point response below.

1. *The text on Figure 1A needs to be enlarged as it's currently illegible.*

Response: Figure 1A with enlarged text is now presented in the revised manuscript.

2. *Presumably, the authors chose the cell lines expressing the highest levels of BMAL2 in Figure 2a to perform the rest of the experiments on? If so, please state this in the text on page 5.*

Response: We thank the reviewer for this comment. We modified and added additional description to this part of the result (page 5). "We examined a panel of OCCC cell lines and found that BMAL2 expression was higher in most of the OCCC cells regardless of the ARID1A status when compared to immortalized human endometrial stromal cells (HESC) and human primary ovarian surface epithelial cells (HOEC), which served as a normal control (Fig. 2A).

OCCC cell lines with high BMAL2 level were used for loss-of-function assays where BMAL2 was depleted with shRNA (Fig. 2B).“

2.-cont. It would have been interesting to see whether the BMAL2 ShRNA could still further inhibit cell growth even in the low expressing cell lines or whether it has no additional effect. Perhaps BMAL2 levels could serve as a biomarker for GW833972A.

Response: We agree with the reviewer that high BMAL2 level could be a good biomarker for GW833972A treatment. To address these comments, RMG1 and TOV21G cell lines were used. Immunoblot failed to detect further depletion of BMAL2 in shBMAL2-transduced cells (Rebuttal Figure 6A) as the endogenous BMAL2 level was very low in these cell lines. Similarly, no further BMAL2 depletion can be detected upon GW833972A treatment of these cells (Rebuttal Fig. 6B, revised Fig. EV7). The GW833972A-treated cells were then subjected to cell viability assay. In TOV21G cells, no significant reduction in cell viability was found when 20 μ M GW833972A was used. RMG1 had a decrease in cell viability when treated with 20 μ M GW833972A, but no significant effect was observed when lower dose was used (Rebuttal Fig. 6C, revised Fig. EV7). These results are consistent with our observations that GW833972A has low adverse effects on normal human fibroblasts (WI38, original Fig. EV6, revised Fig. EV7) and normal tissues in the mouse model (original Fig. EV7, revised Fig. EV8). Thus, we can conclude that depletion of BMAL2 only inhibits cell growth and viability in cells expressing high BMAL2.

3. Many of the blots in the supplementary figures are very blurry and low resolution (this may be an artifact of the uploading process) but these need to be better quality so they can be adequately assessed.

Response: We thank the reviewer for this comment. We think that the low resolution of the EV figures was probably due to the uploading process. We have been careful to upload

a version with high quality images for this resubmission and will carefully check image quality at later stages of manuscript processing.

4. Several of the figure legends state 'The experiments were repeated 3 times.' Please can the authors clarify whether the data presented is representative of the data of the 3 repeats or only one experiment.

Response: The data presented is one representative experiment of three replicates. The description is now clarified in the revised figure legends.

5. BMAL2 is not defined as an essential gene as it's functions are mostly compensated by BMAL1. The authors do touch on BMAL1 levels being low in ovarian cancer and that seems to be the case in their cell lines, but I do wonder if what they are seeing is synthetic lethality in cell lines that are low in BMAL1 already, since BMAL1 and BMAL2 can compensate for each other. The authors should over-express BMAL1 in their cell lines (or select a cell line with high BMAL1 and BMAL2 levels) to assess whether BMAL1 can protect against BMAL2 KD and GW833972A. This should also be addressed in the discussion.

Response: We would like to emphasize that although BMAL2 has structural and functional similarities to BMAL1, the compensatory phenomenon of Bmal2 in Bmal1 knockout mice was only observed when Bmal2 was transgenically overexpressed ((Bunger *et al*, 2000) Cell 103(7): 1009). There is no report that natively expressed BMAL2 can compensate for BMAL1 (or vice versa). More recent studies have demonstrated that BMAL1 and BMAL2 have distinct, or even opposite functions in specific tissues and physiological processes ((Brady *et al*, 2016) Cancer Cell 29(5): 697; (Mandl *et al*, 2022) Cell Death Discov 8(1): 443; (Zhao *et al*, 2023) J Exp Clin Cancer Res 42(1): 229). In ovarian cancer, BMAL1 expression is low and is considered to be a tumor suppressor. BMAL1 low expression is often associated with malignancy and poor prognosis ((Tokunaga *et al*, 2008) Acta Obstet Gynecol Scand 87(10): 1060; (Yeh *et al*, 2014) Int J Oncol 45(5): 2101). This is opposite from the oncogenic role of BMAL2 that we observed.

In the five OCCC cell lines used in our study, ES-2, JHOC5 and OVI5E maintain moderate level of BMAL1 expression, while OVCA429 and JHOC9 have lower BMAL1 expression (Rebuttal Fig. 7, left panel). Take ES-2 cells for example, when BMAL2 was depleted using GW833972A, BMAL1 level did not change (Rebuttal Fig. 7, right panel; revised Fig. EV7). All

Rebuttal Figure 7.

(Left panel) IB of BMAL2 and BMAL1 protein expression with GAPDH as a loading control. Blots shown are from one representative experiment of three replicates.

(Right panel) IB of BMAL2 and BMAL1 protein with GAPDH as a loading control in ES-2 cells treated with vehicle (DMSO), 5, 10 or 20 μM GW833972A. Blots shown are from one representative experiment of three replicates.

five cell lines, regardless of BMAL1 level, showed significantly reduced cell proliferation and viability (Original Fig. 2). These data indicated that BMAL1 does not affect the impact of BMAL2 depletion in OCCC cells. This point is now included in the revised discussion on page 12-13.

6. The authors state that 'BMAL2 knockdown did not lead to increase in γ H2AX foci in HEYA8 and KURAMOCHI cells (Fig. EV2D), suggesting that BMAL2-mediated DNA integrity maintenance is an OCCC subtype-specific regulatory mechanism'. This is a curious observation, but the authors need to show the BMAL1 and BMAL2 levels and extent of the BMAL2 knockdown in the HEYA8 and KURAMOCHI cells to enable a comparison with the OCCC cells. It may just be in tumour cells with high BMAL2 levels (and potentially low BMAL1 levels) rather than specific to OCCC. Therefore, this statement should probably be toned down.

Response: We performed immunoblotting experiments using HEYA8 and KURAMOCHI cells without or with BMAL2 depletion to examine the level of BAML1 and BMAL2. Similar to OCCC, HEYA8 and KURAMOCHI cells express high BMAL2 but moderate to low BMAL1 (Rebuttal Fig. 8, left panel). Transduction of shBMAL2 successfully depleted BMAL2 in these cells (Rebuttal Fig. 8, right panel). Since HEYA8 and KURAMOCHI also express high BMAL2 and low BMAL1, the lack of DNA damage upon BMAL2 depletion in these serous type OC cells (original Fig. EV2D; now in the revised Fig. EV3F) supports our statement that BMAL2-mediated DNA integrity maintenance is an OCCC specific phenomenon. This data is now presented in the revised Fig. EV3.

7. Figure 3D-E, please add the olaparib dose to the figure legend. There is also no figure legend for Figure 3E.

Response: We apologize if the original description in the figure legend left out some details. The dose of Olaparib was described in the main text (page 8). Now this information is added to the revised figure legend. The legend for Figure 3E is now added in the revised manuscript.

8. *A chemist/molecular modelling expert should review the screening section as this is not my primary expertise, but it seems unlikely that 8/9 compounds from the molecular docking screen would affect BMAL2 stability. It is generally estimated that 10-30% of hits from screens will bind and even less will have a functional effect on the protein.*

Response: We thank the reviewer for raising this point. This comment prompted us to revisit the methods section describing the structure-based virtual screening of BMAL2 inhibitor (page 19). We apologize for the following missing information: We used two chemical datasets, T001 (18,100 bioactive compounds and natural products) and LF1000 (50,000 purchasable compounds), to perform the screening. However, in the original manuscript, we only included Dataset T001. This information is now corrected in the revised manuscript (page 19).

We also apologize if our description caused the reviewer to misinterpret our results. Based on the docking scores, drug properties, and binding mode analysis, 119 high-affinity compounds (FRED Chemgauss4 score between -9.3 and -7.8) were identified from chemical Dataset T001 and Dataset LF1000 which include 68,100 compounds. This is approximately 0.17% of hits from the screen. We then selected the top 9 bioactive compounds with low Kd value for further experiments. Thus, the percentage of hits from our screen should be within the reasonable range of molecular docking screen. We added this more detailed information to the revised manuscript in the results section on page 8. “Based on the docking scores, drug properties, and binding mode analysis, 119 high-affinity compounds (FRED Chemgauss4 score between -9.3 and -7.8) were identified from the chemical databases including more than 60,000 compounds. Within the 119 candidates, the top 9 bioactive compounds with low Kd value were then selected for further experiments. “

8-cont. *The authors need to provide the full high-resolution blots as the current blots show very blurry bands and they need to show that the GAPDH and BMAL2 bands are from the same membrane. EV5 should also be a figure in the main manuscript as it is an important part of the manuscript.*

Response: The original immunoblots were replaced by blots with high-resolution. In the revised manuscript, the binding modes of GW833972A to human BMAL2 is moved to the main figure (Fig. 4A). Please also note that after carefully examining this figure, we found that the compound shown in the original Fig. EV5D was incorrect. The correct image for the binding modes of GW833972A to human BMAL2 is now presented in the revised figure. We sincerely apologize for this mistake.

9. *The authors also need to determine whether GW833972A would bind to BMAL1 as the PAS2 domains in BMAL1 and BMAL2 are highly homologous. This could be addressed via modelling (if the BMAL1 structure is available) or the authors could look at BMAL1 protein levels to determine if it is also down regulated.*

Response: We thank the reviewer for this suggestion. The structure of BMAL1 PAS2 (a.k.a. PAS-B) domain in complex with a core circadian modulator (CCM) was recently resolved by X-ray diffraction (Protein Data Bank: 8RW8) (Pu *et al*, 2025). We used this CCM binding pocket to evaluate whether GW833972A binds to BMAL1 PAS2 domain. In contrast to the

surface-exposed binding pocket of BMAL2, the CCM binding site of BMAL1 is an internal cavity (Pu *et al.*, 2025). Also, the residues between BMAL1 and BMAL2 that interact with GW833972A have different physicochemical properties. The residues of the BMAL1 binding pocket are hydrophobic. However, the residues of BMAL2 binding pocket are hydrophilic or uncharged (Rebuttal Fig. 9). Thus, this difference in structure configuration is expected to make it difficult for GW833972A to access the BMAL1 binding pocket. Consistent with this possibility, when BMAL1-expressing cells, such as ES2 and WI38, were treated with GW833972A, BMAL1 protein level was not reduced (Rebuttal Fig. 7 and revised Fig. 7). Together, these results indicate that GW833972A does not target BMAL1 protein. These results and description are now presented in the revised Fig. EV7, the result section (page 10) and the discussion section (page 12-13).

10. *The effect of BMAL2 knockdown or GW833972A on normal human cells has not been explored at all in this study, so targeting BMAL2 could be toxic to non-cancerous human cells. The authors should test BMAL2 knockdown and GW833972A in human primary fibroblasts to assess the effects on cell growth and viability.*

Response: The reviewer has missed this information. The effects of GW833972A on normal human fibroblasts (WI38), including cell viability, DNA damage and dose-response, were described in original Figure EV6A-6C (now in the revised Fig. EV7). The low toxicity of GW833972A in WI38 fibroblasts was likely due to the extremely low BMAL2 expression. Since BMAL2 expression is low in most normal tissues (Fig. 1A), these observations suggest that targeting BMAL2 by GW833972A will not cause significant adverse effects on normal

cells.

11. Discussion: *Since Rad51 inhibitors are already in clinical trials the authors need to discuss how BMAL2 inhibitors may be a different approach, i.e. will they show any advantage over Rad51 inhibitors?*

Response: In terms of molecular mechanism, B02 specifically targets RAD51 to inhibit HR. In contrast, BMAL2 depletion in BMAL2-high OCCC cells significantly downregulates not only RAD51, but also many upstream (such as BRCA1 and RPAs) and downstream (such as BRCA2 and RAD54) key factors in the HR pathway (original Fig. EV3; revised Fig. EV4). BMAL2 depletion also altered expression of genes in cell cycle, DNA replication, cellular senescence and p53 pathways, in addition to HR (Fig. 3A). Together our data show that BMAL2 is upregulated in cancer cells and has extremely low level in normal cells, and show that targeting BMAL2 can impact multiple tumor promoting pathways. Thus, we think that targeting BMAL2 using GW833792A can be a more effective way to eliminate OCCC cells compare to only targeting RAD51 using B02 in combination with DNA-damaging chemotherapeutic drugs, which can cause severe adverse effects in normal cells. We added this part to the revised discussion section on page 13.

Minor typos:

Page 7: Olaprib should be olaparib.

Response: This error is corrected in the revised manuscript.

Referee #3 (Comments on Novelty/Model System for Author):

Experiments were performed in triplicates and appropriate statistical analysis was performed.

Identification of BMAL2 as a druggable oncogene in ARID1A wild-type ovarian clear cell carcinoma is new and medically relevant.

The experiments were performed in vitro (using various OCCC lines) and in mice using xenografts.

Referee #3 (Remarks for Author):

This manuscript focuses on ovarian clear cell carcinoma (OCCC), which has a very poor survival rate and shows chemoresistance, especially in the case of ARID1A wild-type background (50% of OCCC cases). The authors identified BMAL2 as an oncogene in OCCC, which promotes homologous recombination (HR) repair by sustaining the expression of HR genes such as RAD51. BMAL2 is upregulated in ovarian cancer, especially in OCCC. Using structure-based screening, they identified a compound that can block BMAL2. They show that depletion or inhibition of BMAL2 reduces cancer cell proliferation in vitro and in a xenograft model, and that it can synergize with PARP inhibition. The compound does not have the same effects on normal fibroblasts, suggesting that it wouldn't have severe side effects. Overall, the study is medically relevant and well-conducted, the manuscript is clearly written. There are a few missing experiments or explanations that are required to consolidate the overall quality, as indicated below.

Response: We thank the reviewer for the positive feedback. Please find the point-to-point response below.

1. Figure 1F: Please explain at least in the figure legend why PAX8 was used in IHC.

Response: PAX8 is a nuclear marker expressing in epithelial neoplasms of thyroid, thymic, ovarian, endometrial, endocervical, fallopian tube and renal origin (<https://www.pathologyoutlines.com/topic/stainspax8.html>). Studies have shown that 95-100% of OCCC express PAX8, making it a reliable clinical marker for OCCC (Chai *et al*, 2017; Gorbokon *et al*, 2024). In the revised manuscript, we added this information in the figure legend of Fig. 1G (Fig. 1F in the original manuscript).

2. Figure 2A: There is a high variability in BMAL2 expression across OCCC cell lines (some with very high but some with very low BMAL2). Please include a normal ovary cell line and a non-OCCC ovarian cancer cell line for better comparison.

Response: We thank the reviewer for this comment. In the revised Figure 2A, we included immortalized human endometrial stromal cells (HESC) and human primary ovarian surface epithelial cells (HOEC) to the OCCC cell panel and redid the immunoblotting to demonstrate that BMAL2 is upregulated in most of the OCCC cell lines (5 out of 7 cell lines express high BMAL2 level) (Rebuttal Fig. 10). A revised description of this data was also added to the results (page 5): "We examined a panel of OCCC cell lines and found that BMAL2 expression was higher in most of the OCCC cells regardless of the ARID1A status when compared to

immortalized human endometrial stromal cells (HESC) and human primary ovarian surface epithelial cells (HOEC), which served as a normal control (Fig. 2A). OCCC cell lines with high BMAL2 level were used for loss-of-function assays where BMAL2 was depleted with shRNA (Fig. 2B).“

Rebuttal Figure 10. IB of BMAL2 protein with GAPDH as a loading control in immortalized human endometrial stromal cells (HESC), human primary ovarian surface epithelial cells (HOEC) and OCCC cell lines.

We also examined non-OCCC cell lines including a low-grade serous ovarian carcinoma cell line HEYA8, and several commonly used high-grade serous ovarian carcinoma cell lines (KURAMOCHI, OVSAHO, SKOV3, OVCAR3 and OVKATE). Compared to the normal ovarian cells, only HEYA8 and KURAMOCHI expressed higher BMAL2 (2 out of 6 serous cell lines express higher BMAL2, Rebuttal Fig. 8, left panel). This result is consistent with the data in the original Figure 1E (now in the revised Figure 1F) which showed that a high proportion of the OCCC samples had high BMAL2 expression compared to serous subtype. This result is presented in the revised Figure EV3D to support the experiments using HEYA8 and KURAMOCHI to examine whether BMAL2-depletion mediated DNA damage effect differs between OC subtypes.

The reviewer’s comment also prompted us to revise Figure 1 to demonstrate that BMAL2 is upregulated in OC. We added a new result using the CSIOVDB microarray gene expression database to compare BMAL2 expression in normal ovarian surface epithelium (OSE), stroma, fallopian tube epithelium (FTE) and ovarian tumor. A significant higher BMAL2 expression was found in the tumor samples compared to normal tissues (Rebuttal Fig. 1C). This new data is presented in the revised Figure 1B.

3. *Figure 2B-F: RMG1 and TOV21G cell lines should be included as well because they express lower levels of BMAL2, so presumably the effect of BMAL2 depletion on cell proliferation would be weaker. One of these two cell lines should also be tested in mouse experiments (Figure 2G-L).*

Response: To address these comments, RMG1 and TOV21G cell lines were used. Immunoblot failed to detect further depletion of BMAL2 in shBMAL2-transduced cells as the endogenous BMAL2 level was very low in these cell lines (Rebuttal Fig. 6A). Similarly, no further BMAL2 depletion can be detected upon GW833972A treatment of these cells (Rebuttal Fig. 6B). The GW833972A-treated cells were then subjected to cell viability and colony forming assays. In TOV21G cells, no significant reduction in cell viability was found and only a slight decrease in colony forming ability was observed when 20 μ M GW833972A was used. RMG1 had a response to 20 μ M GW833972A, but no significant effect was observed when lower dose was used (Rebuttal Fig. 6C). We also demonstrated that the adverse effect of GW833972A on normal human fibroblasts (WI38) was low (original Fig. EV6; revised Fig. EV7). These results are consistent with our observations that GW833972A

has low adverse effects on normal tissues in the mouse model (original Fig. EV7; revised Fig. EV8).

Since these in vitro experiments correlate well with the mouse xenograft model, we hope the editor and the reviewer agree with us that the experiments described above are sufficient to demonstrate the target specificity of GW833972A in BMAL2-high OCCC cells.

4. Can you please indicate that experiments in Figure 3 and EV2 were performed in untreated cells (without DNA-damaging agents).

Response: This information is added to the figure legend of Figure 3 and revised Fig. EV3 in the revised manuscript.

5. The effect of GW833972A (Figure 5) should be tested in low BMAL2-expressing cell lines like RMG1 and TOV21G to presumably confirm that they would not respond to this compound.

Response: As described in the response to point 3 above, RMG1 and TOV21G cell lines were used. Immunoblot failed to detect further depletion of BMAL2 upon GW833972A treatment of these cells (Rebuttal Fig. 6). The GW833972A-treated cells were then subjected to cell viability and colony forming assays. In TOV21G cells, no significant reduction in cell viability was found and only a slight decrease in colony forming ability was observed when 20 μM GW833972A was used. RMG1 had a response to 20 μM GW833972A, but no significant effect was observed when lower dose was used (Rebuttal Fig. 6; revised Fig. EV7). We also demonstrated that the adverse effect of GW833972A on normal human fibroblasts (WI38) was low (original Fig. EV6; revised Fig. EV7). These results are consistent with our observations that GW833972A has low adverse effects on normal tissues in the mouse model (original Fig. EV7; revised Fig. EV8).

References

- Bolton KL, Chen D, Corona de la Fuente R, Fu Z, Murali R, Kobel M, Tazi Y, Cunningham JM, Chan ICC, Wiley BJ *et al* (2022) Molecular Subclasses of Clear Cell Ovarian Carcinoma and Their Impact on Disease Behavior and Outcomes. *Clin Cancer Res* 28: 4947-4956
- Brady JJ, Chuang CH, Greenside PG, Rogers ZN, Murray CW, Caswell DR, Hartmann U, Connolly AJ, Sweet-Cordero EA, Kundaje A *et al* (2016) An Arntl2-Driven Secretome Enables Lung Adenocarcinoma Metastatic Self-Sufficiency. *Cancer Cell* 29: 697-710
- Bunger MK, Wilsbacher LD, Moran SM, Clendenin C, Radcliffe LA, Hogenesch JB, Simon MC, Takahashi JS, Bradfield CA (2000) Mop3 is an essential component of the master circadian pacemaker in mammals. *Cell* 103: 1009-1017
- Chai HJ, Ren Q, Fan Q, Ye L, Du GY, Du HW, Xu W, Li Y, Zhang L, Cheng ZP (2017) PAX8 is a potential marker for the diagnosis of primary epithelial ovarian cancer. *Oncol Lett* 14: 5871-5875
- Gorbokon N, Baltruschat S, Lennartz M, Luebke AM, Hoflmayer D, Kluth M, Hube-Magg C, Hinsch A, Fraune C, Lebok P *et al* (2024) PAX8 expression in cancerous and non-neoplastic tissue: a tissue microarray study on more than 17,000 tumors from 149 different tumor entities. *Virchows Arch* 485: 491-507
- Mandl M, Viertler HP, Zopoglou M, Mitterberger-Vogt MC, Gasser J, Hatzmann FM, Rauchenwald T, Zwierzina ME, Mattesich M, Weiss AKH *et al* (2022) The circadian transcription factor ARNTL2 is regulated by weight-loss interventions in human white adipose tissue and inhibits adipogenesis. *Cell Death Discov* 8: 443
- Pu H, Bailey LC, Bauer LG, Voronkov M, Baxter M, Huber KVM, Khorasanizadeh S, Ray D, Rastinejad F (2025) Pharmacological targeting of BMAL1 modulates circadian and immune pathways. *Nat Chem Biol* 21: 736-745
- Tokunaga H, Takebayashi Y, Utsunomiya H, Akahira J, Higashimoto M, Mashiko M, Ito K, Niikura H, Takenoshita S, Yaegashi N (2008) Clinicopathological significance of circadian rhythm-related gene expression levels in patients with epithelial ovarian cancer. *Acta Obstet Gynecol Scand* 87: 1060-1070
- Yeh CM, Shay J, Zeng TC, Chou JL, Huang TH, Lai HC, Chan MW (2014) Epigenetic silencing of ARNTL, a circadian gene and potential tumor suppressor in ovarian cancer. *Int J Oncol* 45: 2101-2107
- Zhao D, Dong Y, Duan M, He D, Xie Q, Peng W, Cui W, Jiang J, Cheng Y, Zhang H *et al* (2023) Circadian gene ARNTL initiates circGUCY1A2 transcription to suppress non-small cell lung cancer progression via miR-200c-3p/PTEN signaling. *J Exp Clin Cancer Res* 42: 229

10th Mar 2026

Dear Dr. Hwang-Verslues,

Thank you for submitting your revised study. We have now received the report from the three initial referees, who are satisfied with the revisions. I will therefore be able to accept your manuscript once the following minor editorial matters are addressed:

1/ Referee #1: please address this referee's concern experimentally if feasible, otherwise address this point in the discussion. More generally, please ensure that the discussion focuses on interpreting the new results rather than reiterating the introduction.

2/ Manuscript text:

- Please remove the yellow highlights, and only indicate in track changes mode any new modification.
- Methods/Statistics: please provide a statement on sample size, exclusion/inclusion criteria, blinding and randomization, and fill the author checklist accordingly.
- Author contributions: please remove from the manuscript text. You will be asked to provide CRediT (Contributor Role Taxonomy) terms in the submission system. These replace a narrative author contribution section in the manuscript.

3/ Figures:

- Please note that figure re-use is allowed, but should be mentioned in the figure legends (see for instance Figure 2G, H, I and Figure EV1D, Figure 2B and Figure EV9A).
- Rename Table EV1 to Dataset EV1 and add a legend to the excel file, in a separate tab/worksheet. Adjust the numbering of the EV tables accordingly, including the callouts in the reagents table.
- Remove the list of EV table legends from the manuscript; they should be in the files only .
- Correct the heading to "Expanded View Figure Legends"
- Appendix: Please rename the file with the detailed GW833972A structural characterization "Appendix", add a table of contents, and change the figure nomenclature to "Appendix Figure S1" etc. Please update the heading to "Appendix Supplementary Methods: Detailed GW833972A structural characterization"
- The image on page 20 should be moved to the Appendix.
- Please address the queries from our data editors in the figure legends:
 1. Please indicate the statistical test used for data analysis in the legends of figures 3A, 4E
 2. Please note that the box plots need to be defined in terms of minima, maxima, centre, bounds of box and whiskers, and percentile in the legend of figure 1A
 3. Please note that information related to n is missing in the legends of figures 1A, B, F; 2J-L; 4G, H; 6D, F, G
 4. Please note that the error bars are not defined in the legends of figures 1B, F; 2J-L; 4G, H; 6A, B, G; EV8 B, C

4/ In the author checklist, please fill in:

- the entire section Experimental study design and statistics
- the section on technical vs. biological replicates
- the section on authority granting ethics approval (human participants)
- please check the section on specimen and field samples, as I don't think it applies to your study.

5/ Thank you for providing Source Data. Please make sure that the images provided for Figure 6 have been updated.

6/ Every published paper includes a 'Synopsis' to further enhance discoverability. Synopses are displayed on the journal webpage and are freely accessible to all readers. They include a short stand first (maximum of 300 characters, including space) as well as 2-5 one-sentences bullet points that summarizes the paper. Please write the bullet points to summarize the key NEW findings. They should be designed to be complementary to the abstract - i.e. not repeat the same text. We encourage inclusion of key acronyms and quantitative information (maximum of 30 words / bullet point). Please use the passive voice. Please attach these in a separate file or send them by email, we will incorporate them accordingly.

7/ As part of the EMBO Publications transparent editorial process initiative (see our Editorial at <http://embomolmed.embopress.org/content/2/9/329>), EMBO Molecular Medicine will publish online a Review Process File (RPF) to accompany accepted manuscripts.

This file will be published in conjunction with your paper and will include the anonymous referee reports, your point-by-point response and all pertinent correspondence relating to the manuscript. Let us know whether you agree with the publication of the RPF and as here, if you want to remove or not any figures from it prior to publication.

I look forward to receiving your revised manuscript.

Yours sincerely,

Lise Roth

***** Reviewer's comments *****

Referee #1 (Remarks for Author):

The authors have addressed most of the points raised in the previous review in an appropriate and constructive manner. In particular, I would like to positively evaluate the additional experiments demonstrating the synergistic effects with PARP inhibition, which have now been examined in detail both in vitro and in vivo. These new data substantially strengthen the scientific value of the manuscript.

I also appreciate the authors' clarification that the proposed therapeutic strategy is not intended to be limited to ARID1A-wild-type tumors, but rather is suggested to be applicable regardless of ARID1A status. This point is clearly explained in the rebuttal, and the Kaplan-Meier curves presented in the Extended Figures appropriately reflect this concept. From my perspective, these data can be presented as they are.

One minor point I would like to mention is that, as ovarian clear cell carcinoma is generally considered to be derived from endometrial tissue, comparison of BMAL2 expression with ectopic endometrial tissue (i.e., endometriosis) would further enhance the manuscript. However, I understand that acquisition of such samples may be challenging, and if this represents a practical limitation, it would be reasonable. If feasible, the authors are encouraged to address this point.

Overall, the revised manuscript has been significantly improved and addresses the major concerns raised in the initial review.

Referee #2 (Comments on Novelty/Model System for Author):

The models are appropriate for the study.

Referee #2 (Remarks for Author):

This is a strong study and the authors have adequately addressed my comments, so I am happy to support its publication in EMBO Molecular Medicine.

Referee #3 (Comments on Novelty/Model System for Author):

Experiments were performed in triplicates and appropriate statistical analysis was performed. Identification of BMAL2 as a druggable oncogene in ARID1A wild-type ovarian clear cell carcinoma is new and medically relevant.

The experiments were performed in vitro (using various OCCC lines) and in mice using xenografts.

Referee #3 (Remarks for Author):

The authors adequately addressed my concerns and I now support publication of this revised manuscript.

The authors have addressed all minor editorial requests.

19th Mar 2026

Dear Dr. Hwang-Verslues,

Thank you for submitting your revised files. I am pleased to inform you that your manuscript is accepted for publication and is now being sent to our publisher to be included in the next available issue of EMBO Molecular Medicine.

You may qualify for financial assistance for your publication charges - either via a Springer Nature fully open access agreement or an EMBO initiative. Check your eligibility: <https://link.springer.com/journal/44321/how-to-publish-with-us>

If you have any questions, please do not hesitate to contact the Editorial Office. Thank you for your contribution to EMBO Molecular Medicine!

With kind regards,

Lise

>>> Please note that it is EMBO Molecular Medicine policy for the transcript of the editorial process (containing referee reports and your response letter) to be published as an online supplement to each paper. If you do NOT want this, you will need to inform the Editorial Office via email immediately. More information is available here: <https://link.springer.com/partners/embo-press/editorial-policies#Peer%20review>